# From RAG to Memory: Non-Parametric Continual Learning for Large Language Models

Bernal Jiménez Gutiérrez [* 1]  Yiheng Shu [* 1]  Weijian Qi [1]  Sizhe Zhou [2]  Yu Su [1]

## Abstract

Our ability to continuously acquire, organize, and leverage knowledge is a key feature of human intelligence that AI systems must approximate to unlock their full potential. Given the challenges in continual learning with large language models (LLMs), retrieval-augmented generation (RAG) has become the dominant way to introduce new information. However, its reliance on vector retrieval hinders its ability to mimic the dynamic and interconnected nature of human long-term memory. Recent RAG approaches augment vector embeddings with various structures like knowledge graphs to address some of these gaps, namely sense-making and associativity. However, their performance on more basic factual memory tasks drops considerably below standard RAG. We address this unintended deterioration and propose HippoRAG 2, a framework that outperforms standard RAG comprehensively on factual, sense-making, and associative memory tasks. HippoRAG 2 builds upon the Personalized PageRank algorithm used in HippoRAG and enhances it with deeper passage integration and more effective online use of an LLM. This combination pushes this RAG system closer to the effectiveness of human long-term memory, achieving a 7% improvement in associative memory tasks over the state-of-the-art embedding model while also exhibiting superior factual knowledge and sense-making memory capabilities. This work paves the way for non-parametric continual learning for LLMs. Code and data are available at https://github.com/OSU-NLP-Group/HippoRAG.

[*]Equal contribution  [1]The Ohio State University, Columbus, OH, USA [2]University of Illinois Urbana-Champaign, IL, USA. Correspondence to: Bernal Jiménez Gutiérrez <jimenezgutierrez.1@osu.edu>, Yiheng Shu <shu.251@osu.edu>, Yu Su <su.809@osu.edu>.

*Proceedings of the 42ⁿᵈ International Conference on Machine Learning*, Vancouver, Canada. PMLR 267, 2025. Copyright 2025 by the author(s).

## 1. Introduction

In an ever-evolving world, the ability to continuously absorb, integrate, and leverage knowledge is one of the most important features of human intelligence. From lawyers navigating shifting legal frameworks to researchers tracking multifaceted scientific progress, much of our productivity relies on this incredible capacity for continual learning. It is imperative for AI systems to approximate this capability in order to become truly useful human-level assistants.

In recent years, large language models (LLMs) have made remarkable progress in many aspects of human intelligence. However, efforts to endow these models with our evolving long-term memory capabilities have faced significant challenges in both fully absorbing new knowledge (Zhong et al., 2023; Hoelscher-Obermaier et al., 2023) and avoiding catastrophic forgetting (Cohen et al., 2024; Gu et al., 2024), due to the complex distributional nature of their parametric knowledge. Retrieval-augmented generation (RAG) has emerged as a way to circumvent these obstacles and allow LLMs to access new information in a *non-parametric* fashion without altering an LLM's parametric representation. Due to their simplicity and robustness (Zhong et al., 2023; Xie et al., 2024), RAG has quickly become the *de facto* continual learning solution for production LLM systems. However, their reliance on simple vector retrieval results in the inability to capture two vital aspects of our interconnected long-term memory system: **sense-making** (Klein et al. (2006); the ability to interpret larger, more complex, or uncertain contexts) and **associativity** (Suzuki (2005); the capacity to draw multi-hop connections between disparate pieces of knowledge).

Several RAG frameworks that engage an LLM to explicitly structure its retrieval corpus have been recently proposed to address these limitations. To enhance sense-making, such *structure-augmented* RAG methods allow an LLM to either generate summaries (Edge et al., 2024; Sarthi et al., 2024; Chen et al., 2023) or a knowledge graph (KG) structure (Guo et al., 2024) to link groups of disparate but related passages, thereby improving the RAG system's ability to understand longer and more complex discourse such as long stories. To address the associativity gap, the authors of HippoRAG (Gutiérrez et al., 2024) use the Personalized

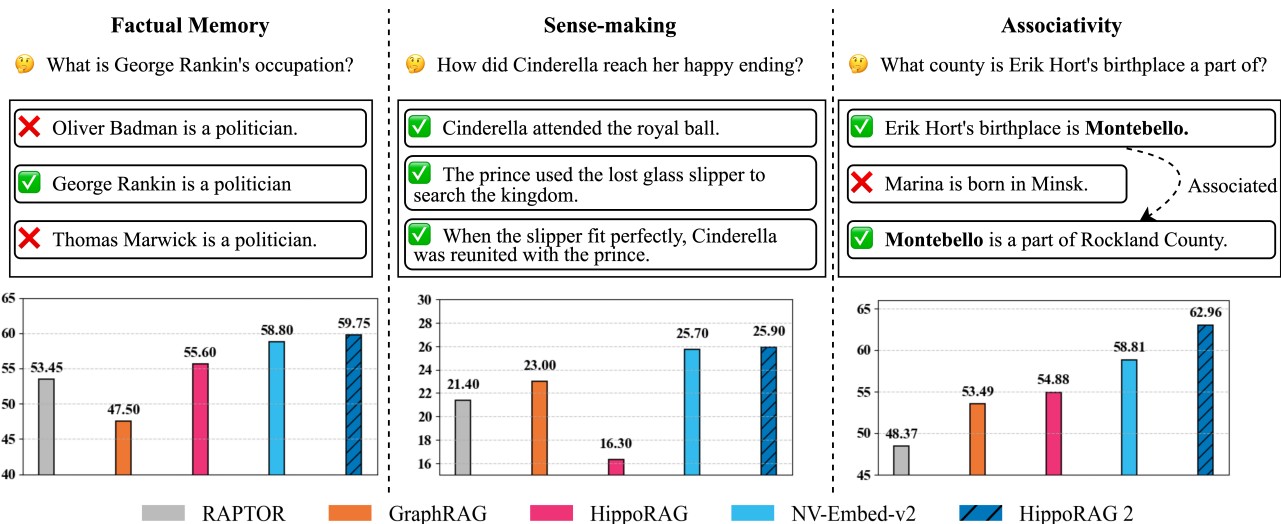

*Figure 1.* Evaluation of continual learning capabilities across three key dimensions: factual memory (NaturalQuestions, PopQA), sense-making (NarrativeQA), and associativity (MuSiQue, 2Wiki, HotpotQA, and LV-Eval). HippoRAG 2 surpasses other methods across all benchmark categories, bringing it one step closer to a true long-term memory system.

PageRank algorithm (Haveliwala, 2002) and an LLM's ability to automatically construct a KG and endow the retrieval process with multi-hop reasoning capabilities.

Although these methods demonstrate strong performance in both of these more challenging memory tasks, bringing RAG truly closer to human long-term memory requires robustness across simpler memory tasks as well. In order to understand whether these systems could achieve such robustness, we conduct comprehensive experiments that not only simultaneously evaluate their associativity and sense-making capacity through multi-hop QA and large-scale discourse understanding, but also test their **factual memory** abilities via simple QA tasks, which standard RAG is already well-equipped to handle.

As shown in Figure 1, our evaluation reveals that all previous structure-augmented methods underperform against the strongest embedding-based RAG methods available on all three benchmark types. Perhaps unsurprisingly, we find that each method type experiences the largest performance decay in tasks outside its own experimental setup. For example, HippoRAG's performance drops most on large-scale discourse understanding due to its lack of query-based contextualization, while RAPTOR's performance deteriorates substantially on the simple and multi-hop QA tasks due to the noise introduced into the retrieval corpora by its LLM summarization mechanism.

In this work, we leverage this experimental setting to help us address the robustness limitations of these innovative approaches while avoiding the pitfalls of focusing too narrowly on just one task. Our proposed method, HippoRAG 2, leverages the strength of HippoRAG's OpenIE and Personalized

PageRank (PPR) methodologies while addressing its query-based contextualization limitations by integrating passages into the PPR graph search process, involving queries more deeply in the selection of KG triples as well as engaging an LLM in the online retrieval process to recognize when retrieved triples are irrelevant.

Through extensive experiments, we find that this design provides HippoRAG 2 with consistent performance improvements over the most powerful standard RAG methods across the board. More specifically, our approach achieves an average 7 point improvement over standard RAG in associativity tasks while showing no deterioration and even slight improvements in factual memory and sense-making tasks. Furthermore, we show that our method is robust to different retrievers as well as to the use of strong open-source and proprietary LLMs, allowing for a wide degree of usage flexibility. All of these results suggest that HippoRAG 2 is a promising step in the development of a more human-like non-parametric continual learning system for LLMs.

## 2. Related Work

### 2.1. Continual Learning for LLMs

As the use of LLMs in real-world applications grows, it becomes increasingly important for them to acquire and integrate new knowledge over time while preserving past information—as evidenced by the many benchmarking efforts in this direction (Zhong et al., 2023; Liska et al., 2022; Kim et al., 2024; Roth et al., 2024; Li et al., 2024). Given the high computational cost of full-scale LLM pretraining, various techniques have been leveraged to endow these mod-

els with this continual learning capacity. These approaches generally fall into three categories: continual fine-tuning, model editing, and RAG (Shi et al., 2024).

**Continual fine-tuning** involves periodically training an LLM on new data. This can be achieved through methods like continual pretraining (Jin et al., 2022), instruction tuning (Zhang et al., 2023), and alignment fine-tuning (Zhang et al., 2024). While effective in incorporating new linguistic patterns and reasoning skills, continual fine-tuning suffers from catastrophic forgetting (Huang et al., 2024), where previously learned knowledge is lost as new data is introduced. Moreover, its computational expense makes frequent updates impractical for real-world applications.

**Model editing** techniques (Yao et al., 2023) provide a more lightweight alternative by directly modifying specific parameters in the model to update its knowledge. However, these updates have been found to be highly localized, having little effect on information associated with the update that should also be changed.

**RAG** has emerged as a scalable and practical alternative for continual learning. Instead of modifying the LLM itself, RAG retrieves relevant external information at inference time, allowing for real-time adaptation to new knowledge. We will discuss several aspects of this non-parametric continual learning solution for LLMs in the next section.

## 2.2. Non-Parametric Continual Learning for LLMs

**Encoder model improvements**, particularly with LLM backbones, have significantly enhanced RAG systems by generating high-quality embeddings that better capture semantic relationships, improving retrieval quality for LLM generation. Recent models (Li et al., 2023; Muennighoff et al., 2025; Lee et al., 2025) leverage LLMs, large corpora, improved architectures, and instruction fine-tuning for notable retrieval gains. NV-Embed-v2 (Lee et al., 2025) serves as the primary comparison in this paper.

**Sense-making** is the ability to understand large-scale or complex events, experiences, or data (Koli et al., 2024). Standard RAG methods are limited in this capacity since they require integrating information from disparate passages, and thus, several RAG frameworks have been proposed to address it. RAPTOR (Sarthi et al., 2024) and GraphRAG (Edge et al., 2024) both generate summaries that integrate their retrieval corpora. However, they follow distinct processes for detecting what to summarize and at what granularity. While RAPTOR uses a Gaussian Mixture Model to detect document clusters to summarize, GraphRAG uses a graph community detection algorithm that can summarize documents, entity clusters with relations, or a combination of these elements. LightRAG (Guo et al., 2024) employs a dual-level retrieval mechanism to enhance comprehensive

information retrieval capabilities in both low-level and high-level knowledge, integrating graph structures with vector retrieval.

Although both GraphRAG and LightRAG use a KG just like our HippoRAG 2 approach, our KG is used to aid in the retrieval process rather than to expand the retrieval corpus itself. This allows HippoRAG 2 to introduce less LLM-generated noise, which deteriorates the performance of these methods in single and multi-hop QA tasks.

**Associativity** is the capacity to draw multi-hop connections between disparate facts for efficient retrieval. It is an important part of continual learning, which standard RAG cannot emulate due to its reliance on independent vector retrieval. HippoRAG (Gutiérrez et al., 2024) is the only RAG framework that has addressed this property by leveraging the PPR algorithm over an explicitly constructed open KG. HippoRAG 2 is closely inspired by HippoRAG, which allows it to perform very well on multi-hop QA tasks. However, its more comprehensive integration of passages, queries, and triples allows it to have a more comprehensive performance across sense-making and factual memory tasks as well.

## 3. HippoRAG 2

### 3.1. Overview

HippoRAG (Gutiérrez et al., 2024) is a neurobiologically inspired long-term memory framework for LLMs in which each component is inspired by its neurobiological analog for human memory. The framework consists of three primary components: 1) an LLM that acts as an artificial neocortex, 2) a KG and the Personalized PageRank algorithm to mirror the auto-associative qualities of the hippocampus and 3) a retrieval encoder that links these two components, reflecting one of the functions of the parahippocampal regions. These components collaborate to replicate the interactions observed in human long-term memory.

HippoRAG's offline indexing process uses an LLM to process passages into KG triples, which are then incorporated into the KG, our artificial hippocampal index. Meanwhile, the retrieval encoder is responsible for detecting synonymy to interconnect information. In HippoRAG's online retrieval process, the LLM neocortex extracts named entities from a query while the retrieval encoder finds their most similar counterparts in the KG. Then, the nodes in the KG corresponding to these entities, which we refer to as seed nodes, are used to run the Personalized PageRank (PPR) algorithm. More specifically, these seed nodes are used to assign the **reset probabilities** within PPR, which alter the original PageRank algorithm to distribute probability towards the seed nodes and their neighborhoods, enabling HippoRAG's context-based retrieval. Although HippoRAG seeks to construct memory from non-parametric RAG, its effectiveness

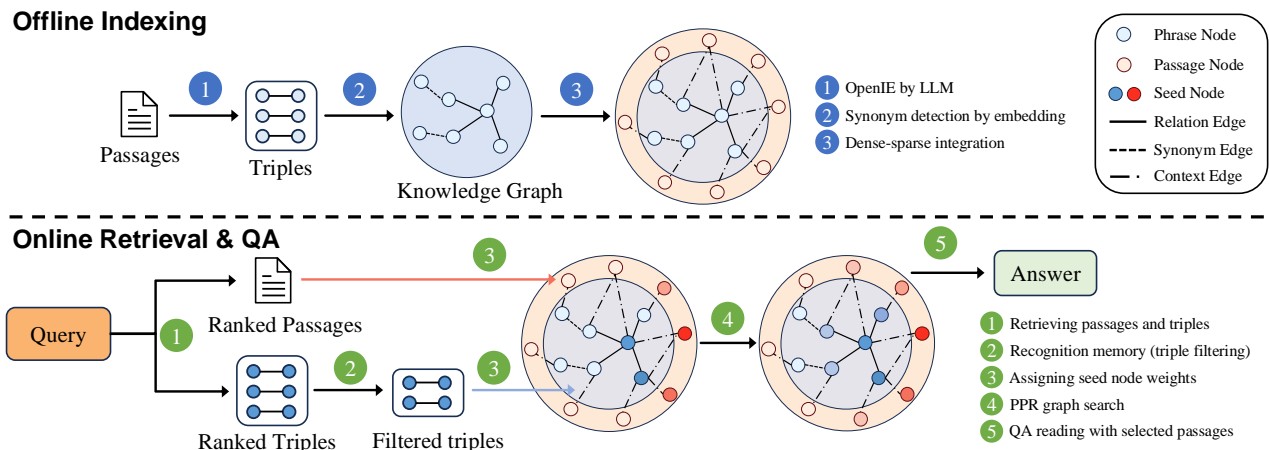

*Figure 2.* **HippoRAG 2 methodology**. For offline indexing, we use an LLM to extract open KG triples from passages, with synonym detection applied to phrase nodes. Together, these phrases and passages form the open KG. For online retrieval, an embedding model scores both the passages and triples to identify the seed nodes of both types for the Personalized PageRank (PPR) algorithm. Recognition memory filters the top triples using an LLM. The PPR algorithm then performs context-based retrieval on the KG to provide the most relevant passages for the final QA task. The different colors shown in the KG nodes above reflect their probability mass; darker shades indicate higher probabilities induced by the PPR process.

is hindered by a critical flaw: an entity-centric approach that causes context loss during both indexing and inference, as well as difficulties in semantic matching.

Built on the neurobiologically inspired long-term memory framework proposed in HippoRAG (Gutiérrez et al., 2024), the structure of HippoRAG 2 follows a similar two-stage process: offline indexing and online retrieval, as shown in Figure 2. Additionally, however, HippoRAG 2 introduces several key refinements that improve its alignment with human memory mechanisms: 1) It seamlessly integrates conceptual and contextual information within the KG, enhancing the comprehensiveness and atomicity of the constructed index (§3.2). 2) It facilitates more context-aware retrieval by leveraging the KG structure beyond isolated KG nodes (§3.3). 3) It incorporates recognition memory to improve seed node selection for graph search (§3.4). In the following sections, we introduce the pipeline in more detail and elaborate on each of these refinements.

**Offline Indexing.** 1) HippoRAG 2, just as HippoRAG, leverages an LLM to extract triples from each passage using OpenIE, which allows the relations and entities to be generated without any constraints or schema. These triples are then arranged into our schema-less KG or hippocampal index. We call the subject or object of these triples **phrases** and the edge connecting them **relation edge**. 2) Next, the retrieval encoder identifies synonyms by evaluating phrase pairs within the KG, detecting those with vector similarity above a predefined threshold, and adding **synonym edge** between such pair. This process enables the KG to link synonyms across different passages, facilitating the integration of both old and new knowledge during learning. 3)

Finally, this phrase-based KG is combined with the original passages, allowing the final open KG to incorporate both conceptual and contextual information (§3.2).

**Online Retrieval.** 1) The query is linked to relevant triples and passages using the encoder, identifying nodes that could be used as seed nodes for graph search (§3.3). 2) During triple linkage, the recognition memory functions as a filter, ensuring only relevant triples are retained from the retrieved set as the final seed nodes (§3.4). 3) These final seed nodes are then used to assign reset probabilities within the PPR algorithm, enabling its context-aware retrieval and refining the linking results to retrieve the most relevant passages. 4) Finally, the retrieved passages serve as contextual inputs for the final QA task. Next, we describe each of the improvements in HippoRAG 2 in more detail.

### 3.2. Dense-Sparse Integration

The nodes in the HippoRAG KG primarily consist of phrases describing concepts, which we refer to as **phrase nodes** in this paper. This graph structure introduces limitations related to the *concept-context tradeoff*. Concepts are concise and easily generalizable but often entail information loss. In contrast, context provide specific circumstances that shape the interpretation and application of these concepts, enriching semantics but increasing complexity. However, in human memory, concepts and contexts are intricately interconnected. The dense and sparse coding theory offers insights into how the brain represents and processes information at different granularities (Beyeler et al., 2019). Dense coding encodes information through the simultaneous activation of many neurons, resulting in a distributed and *redundant*

representation. Conversely, sparse coding relies on minimal neural activation, engaging only a small subset of neurons to enhance *efficiency* and storage *compactness*.

Inspired by the dense-sparse integration observed in the human brain, we treat the phrase node as a form of sparse coding for the extracted concepts, while incorporating dense coding into our KG to represent the context from which these concepts originate. First, we adopt an encoding approach similar to how phrases are encoded, using the embedding model. These two types of coding are then integrated in a specific manner within the KG. Unlike the document ensemble in HippoRAG, which simply aggregates scores from graph search and embedding matching, we enhance the KG by introducing **passage nodes**, enabling more seamless integration of contextual information. This approach retains the same offline indexing process as HippoRAG while enriching the graph structure with additional nodes and edges related to passages during construction. Specifically, each passage in the corpus is treated as a passage node, with the **context edge** labeled "*contains*" connecting the passage to all phrases derived from this passage.

### 3.3. Deeper Contextualization

Building upon the discussion of the concept-context trade-off, we observe that query parsing in HippoRAG, which relies on Named Entity Recognition (NER), is predominantly concept-centric, often overlooking the contextual alignment within the KG. This entity-focused approach to extraction and indexing introduces strong bias toward concepts, leaving many contextual signals underutilized (Gutiérrez et al., 2024). To address this limitation, we explore and evaluate different methods for linking queries to the KG, aiming to more effectively align query semantics with the starting nodes of graph searches. Specifically, we consider three approaches: 1) **NER to Node**: This is the original method used in HippoRAG, where entities are extracted from the query and subsequently matched with nodes in the KG using text embeddings. 2) **Query to Node**: Instead of extracting individual entities, we leverage text embeddings to match the entire query directly to nodes in the KG. 3) **Query to Triple**: To incorporate richer contextual information from the KG, we match the entire query to triples within the graph using text embeddings. Since triples encapsulate fundamental contextual relationships among concepts, this method provides a more comprehensive understanding of the query's intent. By default, HippoRAG 2 adopts the query-to-triple approach, and we evaluate all three methods later (§6.1).

### 3.4. Recognition Memory

Recall and recognition are two complementary processes in human memory retrieval (Uner & Roediger III, 2022). Recall involves actively retrieving information without external cues, while recognition relies on identifying information with the help of external stimuli. Inspired by this, we model the query-to-triple retrieval as a two-step process. 1) **Query to Triple**: We use the embedding model to retrieve the top-k triples $T$ of the graph as described in §3.3. 2) **Triple Filtering**: We use LLMs to filter retrieved $T$ and generate triples $T' \subseteq T$. The detailed prompts are shown in Appendix A.

### 3.5. Online Retrieval

We summarize the online retrieval process in HippoRAG 2 after introducing the above improvements. The task involves selecting seed nodes and assigning reset probabilities for retrieval. HippoRAG 2 identifies phrase nodes from filtered triples generated by query-to-triple and recognition memory. If no triples are available, it directly retrieves top-ranked passages using the embedding model. Otherwise, up to $k$ phrase nodes are selected based on their average ranking scores across filtered triples they originate. All passage nodes are also taken as seed nodes, as broader activation improves multi-hop reasoning. Reset probabilities are assigned based on ranking scores for phrase nodes, while passage nodes receive scores proportional to their embedding similarity, adjusted by a weight factor (§6.2) to balance the influence between phrase nodes and passage nodes. The PPR search is then executed, and passages are ranked by their PageRank scores, with the top-ranked passages used for downstream QA. An example of the pipeline is in Appendix B and the PPR initialization is detailed in Appendix G.1,

## 4. Experimental Setup

### 4.1. Baselines

We select three different types of baselines for comparison. We include three simple baselines: the classic **BM25** (Robertson & Walker, 1994) baseline as well as **Contriever** (Izacard et al., 2022) and **GTR** (Ni et al., 2022), two popular dense embedding retrievers.

Our second baseline category includes some of the largest embedding models available (7B) that demonstrate strong performance on the BEIR leaderboard (Thakur et al., 2021): **Alibaba-NLP/GTE-Qwen2-7B-Instruct** (Li et al., 2023), **GritLM/GritLM-7B** (Muennighoff et al., 2025), and **nvidia/NV-Embed-v2** (Lee et al., 2025).

In our final baseline category, we include four structure-augmented RAG methods. **RAPTOR** (Sarthi et al., 2024) organizes the retrieval corpus into a hierarchical structure based on semantic similarity. **GraphRAG** (Edge et al., 2024) and **LightRAG** (Guo et al., 2024) leverage a KG structure like ours to generate high-level summaries of the concepts present in the corpus. Finally, **HippoRAG** (Gutiérrez et al., 2024) uses a KG as well but integrates knowledge using PPR rather than summarization.

*Table 1.* Dataset statistics

|  | NQ | PopQA | MuSiQue | 2Wiki | HotpotQA | LV-Eval | NarrativeQA |
|---|---|---|---|---|---|---|---|
| Num of queries | $1,000$ | $1,000$ | $1,000$ | $1,000$ | $1,000$ | $124$ | $293$ |
| Num of passages | $9,633$ | $8,676$ | $11,656$ | $6,119$ | $9,811$ | $22,849$ | $4,111$ |

## 4.2. Datasets

To evaluate how well RAG systems retain factual memory while enhancing associativity and sense-making, we select datasets that correspond to three critical challenge types.

1. **Simple QA** primarily evaluates the ability to recall and retrieve factual knowledge accurately.

2. **Multi-hop QA** measures associativity by requiring the model to connect multiple pieces of information to derive an answer.

3. **Discourse understanding** evaluates sense-making by testing the capability to interpret and reason over lengthy, complex narratives.

We will now list the datasets chosen for each category and describe them in detail. The statistics for our sampled datasets are summarized in Table 1.

**Simple QA.** This common type of QA task primarily involves questions centered around individual entities, making it particularly well-suited for embedding models to retrieve relevant contextual information intuitively. We randomly collect $1,000$ queries from the **NaturalQuestions** (NQ) dataset (collected by Wang et al. (2024)), which contains real user questions with a wide range of topics. Additionally, we select $1,000$ queries from **PopQA** (Mallen et al., 2023), with the corpus derived from the December 2021 Wikipedia dump.[1] Both datasets offer straightforward QA pairs, enabling evaluation of single-hop QA capabilities in RAG systems. Notably, PopQA from Wikipedia is especially entity-centric, with entities being less frequent than NaturalQuestions, making it an excellent resource for evaluating entity recognition and retrieval in simple QA tasks.

**Multi-hop QA.** We randomly collect $1,000$ queries from **MuSiQue**, **2WikiMultihopQA**, and **HotpotQA** following HippoRAG (Gutiérrez et al., 2024), all requiring multi-passage reasoning. Additionally, we include all 124 queries from **LV-Eval** (hotpotwikiqa-mixup 256k) (Yuan et al., 2024), a challenging dataset designed to minimize knowledge leakage and reduce overfitting through keyword and phrase replacements. Thus, unlike Wikipedia-based datasets, LV-Eval better evaluates the model's ability to synthesize knowledge from different sources effectively. For

corpus collection, we segment long-form contexts of LV-Eval into shorter passages while maintaining the same RAG setup as other multi-hop datasets.

**Discourse Understanding.** This category consists of only **NarrativeQA**, a QA dataset that contains questions requiring a cohesive understanding of a full-length novel. This dataset's focus on large-scale discourse understanding allows us to leverage it in our evaluation of sense-making in our chosen baselines and our own method. We randomly select 10 lengthy documents and their corresponding 293 queries from NarrativeQA and collect a retrieval corpus just as in the above LV-Eval dataset.

## 4.3. Metrics

Following HippoRAG (Gutiérrez et al., 2024), we use passage recall@5 to evaluate the retrieval task. For the QA task, we follow evaluation metrics from MuSiQue (Trivedi et al., 2022) to calculate token-based F1 scores.

## 4.4. Implementation Details

For HippoRAG 2, we use the open-source Llama-3.3-70B-Instruct (AI@Meta, 2024) as both the extraction (NER and OpenIE) and triple filtering model, and we use nvidia/NV-Embed-v2 as the retriever. We also reproduce the compared structure-augmented RAG methods using the same extractor and retriever for a fair comparison. For the triple filter, we use DSPy (Khattab et al., 2024) MIPROv2 optimizer and Llama-3.3-70B-Instruct to tune the prompt, including the instructions and demonstrations. The resulting prompt is shown in Appendix A. We use top-5 triples ranked by retriever for filtering. Our QA module uses the top-5 retrieved passages as context for an LLM (GPT-4o-mini or Llama-3.3-70B-Instruct) to generate the final answer. For hyperparameters, we follow the default settings from HippoRAG. More implementation and hyperparameter details can be found in Appendix G.

## 5. Results

We now present our main QA and retrieval experimental results, where the QA process uses retrieved results as its context. More detailed experimental results are presented in Appendix C. The statistics for all constructed KGs are shown in Appendix A.

---

[1]https://github.com/facebookresearch/atlas?tab=readme-ov-file#corpora

*Table 2.* **QA performance** (F1 scores) on RAG benchmarks using Llama-3.3-70B-Instruct as the QA reader. No retrieval means evaluating the parametric knowledge of the readers. All structure-augmented RAG baselines and HippoRAG 2 use Llama-3.3-70B-Instruct as the LLM to generate their structure and NV-Embed-v2 as their retriever. This table, along with the following ones, highlight the **best** and second-best results. A bootstrapped statistical test was used to assess significance; [†] indicates that HippoRAG 2 significantly outperforms the best NV-Embed-v2 baseline ($p < 0.05$).

| | Simple QA | | Multi-Hop QA | | | | Discourse Understanding | |
|---|---|---|---|---|---|---|---|---|
| Retrieval | NQ | PopQA | MuSiQue | 2Wiki | HotpotQA | LV-Eval | NarrativeQA | Avg |
| *Simple Baselines* | | | | | | | | |
| None | 54.9 | 32.5 | 26.1 | 42.8 | 47.3 | 6.0 | 12.9 | 38.4 |
| Contriever (Izacard et al., 2022) | 58.9 | 53.1 | 31.3 | 41.9 | 62.3 | 8.1 | 19.7 | 46.9 |
| BM25 (Robertson & Walker, 1994) | 59.0 | 49.9 | 28.8 | 51.2 | 63.4 | 5.9 | 18.3 | 47.7 |
| GTR (T5-base) (Ni et al., 2022) | 59.9 | 56.2 | 34.6 | 52.8 | 62.8 | 7.1 | 19.9 | 50.4 |
| *Large Embedding Models* | | | | | | | | |
| GTE-Qwen2-7B-Instruct (Li et al., 2023) | 62.0 | **56.3** | 40.9 | 60.0 | 71.0 | 7.1 | 21.3 | 54.9 |
| GritLM-7B (Muennighoff et al., 2025) | 61.3 | 55.8 | 44.8 | 60.6 | 73.3 | 9.8 | 23.9 | 56.1 |
| NV-Embed-v2 (7B) (Lee et al., 2025) | 61.9 | 55.7 | 45.7 | 61.5 | 75.3 | 9.8 | 25.7 | 57.0 |
| *Structure-Augmented RAG* | | | | | | | | |
| RAPTOR (Sarthi et al., 2024) | 50.7 | 56.2 | 28.9 | 52.1 | 69.5 | 5.0 | 21.4 | 48.8 |
| GraphRAG (Edge et al., 2024) | 46.9 | 48.1 | 38.5 | 58.6 | 68.6 | 11.2 | 23.0 | 49.6 |
| LightRAG (Guo et al., 2024) | 16.6 | 2.4 | 1.6 | 11.6 | 2.4 | 1.0 | 3.7 | 6.6 |
| HippoRAG (Gutiérrez et al., 2024) | 55.3 | 55.9 | 35.1 | **71.8** | 63.5 | 8.4 | 16.3 | 53.1 |
| HippoRAG 2 | **63.3**[†] | 56.2 | **48.6**[†] | 71.0[†] | **75.5** | **12.9**[†] | **25.9** | **59.8** |

*Table 3.* **Retrieval performance** (passage recall@5) on RAG benchmarks. * denotes the report from the original paper. The compared structure-augmented RAG methods are reproduced with the same LLM and retriever as ours for a fair comparison. GraphRAG and LightRAG are not presented because they do not directly produce passage retrieval results.

| | Simple QA | | Multi-Hop QA | | | |
|---|---|---|---|---|---|---|
| Retrieval | NQ | PopQA | MuSiQue | 2Wiki | HotpotQA | Avg |
| *Simple Baselines* | | | | | | |
| BM25 (Robertson & Walker, 1994) | 56.1 | 35.7 | 43.5 | 65.3 | 74.8 | 55.1 |
| Contriever (Izacard et al., 2022) | 54.6 | 43.2 | 46.6 | 57.5 | 75.3 | 55.4 |
| GTR (T5-base) (Ni et al., 2022) | 63.4 | 49.4 | 49.1 | 67.9 | 73.9 | 60.7 |
| *Large Embedding Models* | | | | | | |
| GTE-Qwen2-7B-Instruct (Li et al., 2023) | 74.3 | 50.6 | 63.6 | 74.8 | 89.1 | 70.5 |
| GritLM-7B (Muennighoff et al., 2025) | 76.6 | 50.1 | 65.9 | 76.0 | 92.4 | 72.2 |
| NV-Embed-v2 (7B) (Lee et al., 2025) | 75.4 | 51.0 | 69.7 | 76.5 | 94.5 | 73.4 |
| *Structure-Augmented RAG* | | | | | | |
| RAPTOR (Sarthi et al., 2024) | 68.3 | 48.7 | 57.8 | 66.2 | 86.9 | 65.6 |
| HippoRAG* (Gutiérrez et al., 2024) | – | – | 51.9 | 89.1 | 77.7 | – |
| HippoRAG (reproduced) | 44.4 | **53.8** | 53.2 | **90.4** | 77.3 | 63.8 |
| HippoRAG 2 | **78.0** | 51.7 | **74.7** | **90.4** | **96.3** | **78.2** |

**QA Performance.** Table 2 presents the QA performance of various retrievers across multiple RAG benchmarks using Llama-3.3-70B-Instruct as the QA reader. HippoRAG 2 achieves the highest average F1 score, demonstrating robustness across different settings. Large embedding models outperform smaller ones, with NV-Embed-v2 (7B) scoring 6.6% higher on average than GTR (T5-base). These models also surpass structure-augmented RAG methods with lower computational costs but excel mainly in simple QA while struggling in complex cases. Notably, HippoRAG 2 outperforms NV-Embed-v2 by 9.5% F1 on 2Wiki and by 3.1% on the challenging LV-Eval dataset. Compared to Hip-

poRAG, HippoRAG 2 shows even greater improvements, validating its neuropsychology-inspired approach. These results highlight HippoRAG 2 as a state-of-the-art RAG system that enhances both retrieval and QA performance while being effectively powered by an open-source model. Table 8 in Appendix C presents additional QA results (EM and F1) using Llama or GPT-4o-mini as the QA reader, along with an extractor or triple filter. GPT-4o-mini follows Llama's trend, with NV-Embed-v2 outperforming structure-augmented methods in most cases, except for HippoRAG in multi-hop QA. HippoRAG 2 consistently outperforms all other methods across nearly all settings. An analysis

*Table 4.* **Ablations.** We report passage recall@5 on multi-hop QA benchmarks using several alternatives to our final design in graph linking, graph construction and triple filtering.

|  | MuSiQue | 2Wiki | HotpotQA | Avg |
|---|---|---|---|---|
| HippoRAG 2 | **74.7** | 90.4 | **96.3** | **87.1** |
| w/ NER to node | 53.8 | **91.2** | 78.8 | 74.6 |
| w/ Query to node | 44.9 | 65.5 | 68.3 | 59.6 |
| w/o Passage Node | 63.7 | 90.3 | 88.9 | 81.0 |
| w/o Filter | 73.0 | 90.7 | 95.4 | 86.4 |

*Table 5.* **Reset probability factor.** Passage recall@5 with different weight factors for passage nodes on our MuSiQue dev set and NaturalQuestions (NQ) dev set, where each set has $1,000$ queries.

| Weight | 0.01 | 0.05 | 0.1 | 0.3 | 0.5 |
|---|---|---|---|---|---|
| MuSiQue | 79.9 | **80.5** | 79.8 | 78.4 | 77.9 |
| NQ | 75.6 | **76.9** | 76.9 | 76.7 | 76.4 |

of the computational resources (tokens, time and memory) required for each method can be found in Appendix F.

**Retrieval Performance.** We report retrieval results for datasets with supporting passage annotations and models that explicitly retrieve passages in Table 3. Large embedding models (7B) significantly outperform classic smaller LM-based models like Contriever and GTR, achieving at least a $9.8\%$ higher F1 score. While our reproduction of HippoRAG using Llama-3.3-70B-Instruct and NV-Embed-v2 shows slight improvements over the original paper, the gains are minimal, with only a $1.3\%$ increase in F1. Although HippoRAG excels in entity-centric retrieval, achieving the highest recall@5 on PopQA, it generally lags behind recent dense retrievers and HippoRAG 2. Notably, HippoRAG 2 achieves the highest recall scores across most datasets, with substantial improvements of $5.0\%$ and $13.9\%$ in Recall@5 on MuSiQue and 2Wiki, respectively, compared to the strongest dense retriever, NV-Embed-v2.

# 6. Discussions

## 6.1. Ablation Study

We design ablation experiments for the proposed linking method, graph construction method, and triple filtering method, with the results reported in Table 4. Each introduced mechanism boosts HippoRAG 2. First, the linking method with deeper contextualization leads to significant performance improvements. Notably, we do not apply a filtering process to the NER-to-node or query-to-node methods; however, the query-to-triple approach, regardless of whether filtering is applied, consistently outperforms the other two linking strategies. On average, query-to-triple improves Recall@5 by $12.5\%$ compared to NER-to-node.

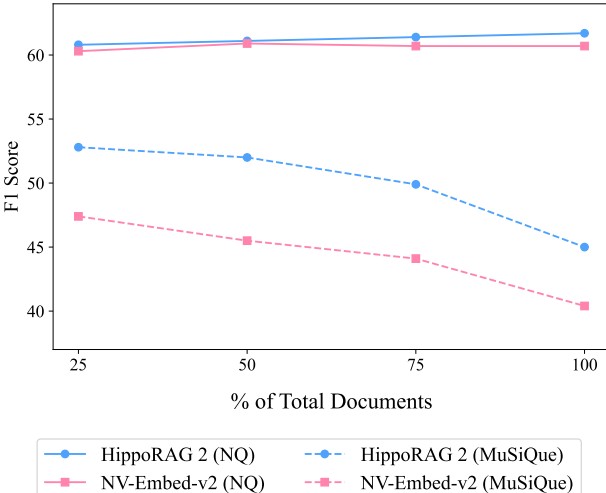

*Figure 3.* **Continual learning experiment:** We partition the NQ and MuSiQue datasets into 4 segments and report the F1 score on a randomly chosen segment as the other 3 segments are introduced into the retrieval corpus to simulate a continuously evolving corpus.

Moreover, query-to-node does not provide an advantage over NER-to-node, as queries and KG nodes operate at different levels of granularity, whereas both NER results and KG nodes correspond to phrase-level representations.

## 6.2. Controlling Reset Probabilities

When setting the reset probability before starting PPR, we find that it is necessary to balance the reset probabilities between two types of nodes: phrase nodes and passage nodes. Specifically, the reset probability of all passage nodes is multiplied by a weight factor to balance the importance of two types of nodes during PPR. Here, we present the results obtained on the validation set in Table 5, which shows that this factor is crucial for the PPR results. Considering the model performance across different scenarios, we set the factor to be $0.05$ by default.

## 6.3. Robustness to Corpus Expansion

As RAG systems become more widely adopted in the real-world, they must increasingly adapt to continual learning scenarios in which the retrieval corpora grow continuously. To understand how HippoRAG 2's capacity to handle this setting compared to standard RAG, we design an experiment in which we partition NQ and MuSiQue into four equal segments, each containing the gold documents and distractors for approximately 250 questions. We then select one segment for evaluation and incrementally add the remaining segments, measuring how performance evolves as new knowledge is added, allowing us to simulate a continual learning setting. We show the F1 score for HippoRAG 2 and NV-Embed-v2, our strongest baseline, in Figure 3.

*Table 6.* We show exemplary retrieval results (the title of passages) from HippoRAG 2 and NV-Embed-v2 on different types of questions. Bolded items denote the titles of supporting passages.

|  | Question | NV-Embed-v2 Results | HippoRAG 2 Filtered Triples | HippoRAG 2 Results |
|---|---|---|---|---|
| **Simple QA** | In what city was I.P. Paul born? | **1. I. P. Paul**
2. Yinka Ayefele - Early life
3. Paul Parker (singer) | (**I. P. Paul**, from, **Thrissur**)
(**I. P. Paul**, was mayor of, Thrissur municipal corporation) | **1. I. P. Paul**
**2. Thrissur**
3. Yinka Ayefele |
| **Multi-Hop QA** | What county is Erik Hort's birthplace a part of? | **1. Erik Hort**
2. Horton Park (Saint Paul, Minnesota)
3. Hertfordshire | (**Erik Hort**, born in, **Montebello**)
(**Erik Hort**, born in, New York) | **1. Erik Hort**
2. Horton Park (Saint Paul, Minnesota)
**3. Monstebello, New York** |

*Table 7.* **Robust to different dense retrievers.** Passage recall@5 on MuSiQue subset.

| Retriever | Dense Retrieval | HippoRAG 2 |
|---|---|---|
| GTE-Qwen2-7B-Instruct | 63.6 | **68.8** |
| GritLM-7B | 66.0 | **71.6** |
| NV-Embed-v2 (7B) | 69.7 | **74.7** |

As we can see in Figure 3, HippoRAG 2's improvements over NV-Embed-v2 remain remarkably consistent in both simple (NQ) and associative (MuSiQue) continual learning settings. We also note that, while both methods retain strong performance on simple QA (solid lines) as more knowledge is introduced, their performance in the more complex associative task (dotted lines) degrades at a similar rate as more information is introduced. This divergence underscores the importance of incorporating varied task complexities into future continual learning benchmarks.

### 6.4. Dense Retriever Flexibility

As demonstrated in Table 7, HippoRAG 2 consistently surpasses direct dense retrieval across various retrievers. Notably, these performance gains remain robust regardless of the specific dense retriever used.

### 6.5. Qualitative Analysis

We show examples from PopQA and MuSiQue in Table 6. For the first example, "*In what city was I. P. Paul born?*", NV-Embed-v2 ranks the entity mentioned in the query "*I. P. Paul*" as the top 1, where the passage is enough to answer this question. But HippoRAG 2 does even better. It directly finds the answer "*Thrissur*" when linking the triples, and during the subsequent graph search, it places the passage corresponding to that entity in the second position, which is a perfect retrieval result. For the second multi-hop question, "*What county is Erik Hort's birthplace a part of?*" NV-Embed-v2 also easily identifies the person mentioned, "*Erik Hort.*" However, since this question requires two-step reasoning, it is not sufficient to fully answer the question. In contrast, HippoRAG 2 retrieves a passage titled

"*Montebello*" during the query-to-triple step, which contains geographic information that implies the answer to the question. In the subsequent graph search, this passage is also ranked at the top. Apart from this, the error analysis of HippoRAG 2 is detailed in Appendix E.

## 7. Conclusion

We introduced HippoRAG 2, a novel framework designed to address the limitations of existing RAG systems in approximating the dynamic and interconnected nature of human long-term memory. It combining the strengths of the Personalized PageRank algorithm, deeper passage integration, and effective online use of LLMs. HippoRAG 2 opens new avenues for research in continual learning and long-term memory for LLMs by achieving comprehensive improvements over standard RAG methods across factual, sense-making, and associative memory tasks, showing capabilities that previous methods have either overlooked or been incapable of achieving in a thorough evaluation. Future work could consider leveraging graph-based retrieval methods to further enhance the episodic memory capabilities of LLMs in long conversations.

## Impact Statement

This paper presents work on Retrieval-Augmented Generation (RAG) to advance the field of long-term memory for large language models. While our work may have various societal implications, we do not identify any concerns that warrant specific emphasis beyond those generally associated with large language models and information retrieval systems.

## Acknowledgments

We would also like to extend our appreciation to colleagues from the OSU NLP group for their constructive comments. This work is supported in part by ARL W911NF2220144, NSF 2112606, and a gift from Cisco. We also thank the Ohio Supercomputer Center for providing computational resources. The views and conclusions contained herein

are those of the authors and should not be interpreted as representing the official policies, either expressed or implied, of the U.S. government. The U.S. government is authorized to reproduce and distribute reprints for government purposes notwithstanding any copyright notice herein.

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

# Appendices

Within this supplementary material, we elaborate on the following aspects:

- Appendix A: LLM Prompts

- Appendix B: HippoRAG 2 Pipeline Example

- Appendix C: Detailed Experimental Results

- Appendix D: Graph Statistics

- Appendix E: Error Analysis

- Appendix F: Cost and Efficiency

- Appendix G: Implementation Details and Hyperparameters

## A. LLM Prompts

We show LLM prompts for triple filter in Figure 4, including the instruction, the few-shot demonstrations and the input format.

## B. Pipeline Example

We show a pipeline example of HippoRAG 2 online retrieval in Figure 5, including query-to-triple, triple filtering and using seed nodes for PPR.

## C. Detailed Experimental Results

We show QA performance and retrieval performance with the proprietary model GPT-4o-mini as well as more metrics here, as shown in Table 8 and Table 9.

**QA Performance** As shown in Table 8, when using GPT-4o-mini for indexing and QA reading, HippoRAG 2 consistently achieves competitive EM and F1 scores across most datasets. Notably, it leads in the MuSiQue and 2Wiki benchmarks. Our method also demonstrates superior performance in the NarrativeQA and LV-Eval tasks. When compared to the strong NV-Embed-v2 retriever, HippoRAG 2 exhibits comparable or enhanced F1 scores, particularly excelling in the LV-Eval dataset with reduced knowledge leakage.

**Retrieval Performance** As shown in Table 9, the improvement trend of HippoRAG 2 in recall@2 is similar to that in recall@5.

## D. Graph Statistics

We show the knowledge graph statistics using Llama-3.3-70B-Instruct or GPT-4o-mini for OpenIE in Table 10.

## E. Error Analysis

We provide an error analysis of 100 samples generated by HippoRAG 2 with recall@5 less than 1.0. Among these samples, 26%, 41%, and 33% are classified as 2-hop, 3-hop, and 4-hop questions, respectively. Triple filtering and the graph search algorithm are the two main sources of errors.

**Recognition Memory** In 7% of the samples, no phrase from the supporting documents is matched with the phrases obtained by the query-to-triple stage before triple filtering. In 26% of the samples, no phrase from the supporting documents is matched with the phrases after triple filtering. After the triple filtering step, 8% of the samples show a decrease in the proportion of phrases in the triples that match phrases from the supporting passages. For instance, the first case from Table 11 shows an empty list after triple filtering, which eliminates all relevant phrases. Additionally, 18% of the samples are left with zero triples after filtering. Although not necessarily an error in filtering, this indicates that the attempt to link to

## Triple Filter

**Instruction:**

You are a critical component of a high-stakes question-answering system used by top researchers and decision-makers worldwide. Your task is to filter facts based on their relevance to a given query, ensuring that the most crucial information is presented to these stakeholders. The query requires careful analysis and possibly multi-hop reasoning to connect different pieces of information.
You must select up to 4 relevant facts from the provided candidate list that have a strong connection to the query, aiding in reasoning and providing an accurate answer.
The output should be in JSON format, e.g., {"fact": [["s1", "p1", "o1"], ["s2", "p2", "o2"]]}, and if no facts are relevant, return an empty list, {"fact": []}.
The accuracy of your response is paramount, as it will directly impact the decisions made by these high-level stakeholders. You must only use facts from the candidate list and not generate new facts. The future of critical decision-making relies on your ability to accurately filter and present relevant information.

**Demonstration:**

**Question**: Are Imperial River (Florida) and Amaradia (Dolj) both located in the same country?
**Fact Before Filter**: "{"fact": [["imperial river", "is located in", "florida"], ["imperial river", "is a river in", "united states"], ["imperial river", "may refer to", "south america"], ["amaradia", "flows through", "ro ia de amaradia"], ["imperial river", "may refer to", "united states"]]}",
**Fact After Filter**: "{"fact":[["imperial river","is located in","florida"],["imperial river","is a river in","united states"],["amaradia","flows through","ro ia de amaradia"]]}"

**Question**: When is the director of film The Ancestor 's birthday?
**Fact Before Filter**: "{"fact": [["jean jacques annaud", "born on", "1 october 1943"], ["tsui hark", "born on", "15 february 1950"], ["pablo trapero", "born on", "4 october 1971"], ["the ancestor", "directed by", "guido brignone"], ["benh zeitlin", "born on", "october 14 1982"]]}
**Fact After Filter**: "{"fact":[["the ancestor","directed by","guido brignone"]]}"

**Question**: In what geographic region is the country where Teafuone is located?
**Fact Before Filter**: "{"fact": [["teafuaniua", "is on the", "east"], ["motuloa", "lies between", "teafuaniua"], ["motuloa", "lies between", "teafuanonu"], ["teafuone", "is", "islet"], ["teafuone", "located in", "nukufetau"]]}"
**Fact After Filter**: "{"fact":[["teafuone","is","islet"],["teafuone","located in","nukufetau"]]}"

**Question**: When did the director of film S.O.B. (Film) die?
**Fact Before Filter**: "{"fact": [["allan dwan", "died on", "28 december 1981"], ["s o b", "written and directed by", "blake edwards"], ["robert aldrich", "died on", "december 5 1983"], ["robert siodmak", "died on", "10 march 1973"], ["bernardo bertolucci", "died on", "26 november 2018"]]}
**Fact After Filter**: "{"fact":[["s o b","written and directed by","blake edwards"]]}"

**Question**: Do both films: Gloria (1980 Film) and A New Life (Film) have the directors from the same country?
**Fact Before Filter**: "{"fact": [["sebasti n lelio watt", "received acclaim for directing", "gloria"], ["gloria", "is", "1980 american thriller crime drama film"], ["a brand new life", "is directed by", "ounie lecomte"], ["gloria", "written and directed by", "john cassavetes"], ["a new life", "directed by", "alan alda"]]}
**Fact After Filter**: "{"fact":[["gloria","is","1980 american thriller crime drama film"],["gloria","written and directed by","john cassavetes"],["a new life","directed by","alan alda"]]}"

**Question**: What is the date of death of the director of film The Old Guard (1960 Film)?
**Fact Before Filter**: "{"fact": [["the old guard", "is", "1960 french comedy film"], ["gilles grangier", "directed", "the old guard"], ["the old guard", "directed by", "gilles grangier"], ["the old fritz", "directed by", "gerhard lamprecht"], ["oswald albert mitchell", "directed", "old mother riley series of films"]]}
**Fact After Filter**: "{"fact":[["the old guard","is","1960 french comedy film"],["gilles grangier","directed","the old guard"],["the old guard","directed by","gilles grangier"]]}"

**Question**: When is the composer of film Aulad (1968 Film) 's birthday?
**Fact Before Filter**: "{"fact": [["aulad", "has music composed by", "chitragupta shrivastava"], ["aadmi sadak ka", "has music by", "ravi"], ["ravi shankar sharma", "composed music for", "hindi films"], ["gulzar", "was born on", "18 august 1934"], ["aulad", "is a", "1968 hindi language drama film"]]}
**Fact After Filter**: "{"fact":[["aulad","has music composed by","chitragupta shrivastava"],["aulad","is a","1968 hindi language drama film"]]}"

**Input:**

Question: {}
Fact Before Filter: {}
Fact After Filter: {}

*Figure 4.* LLM prompts for triple filtering (recognition memory).

---

### Query to Triple

**Question**    What county is Erik Hort's birthplace a part of?
**Triples**    ("Erik Hort", "born in", "Montebello"), ("Erik Hort", "born in", "New York"), ("Erik Hort", "is a", "American"), ("Erik Hort", "born on", "February 16, 1987"), ("Erik Hort", "is a", "Soccer player")
**Filtered Triples**    ("Erik Hort", "born in", "Montebello"), ("Erik Hort", "born in", "New York")

---

### PPR Seed Nodes

**Seed Phrase Nodes**    ("Montebello", 1.0), ("Erik Hort", 0.995), ("New York", 0.989)
**Seed Passage Nodes (Title)**    ("Erik Hort", 0.05), ("Horton Park (Saint Paul, Minnesota)", 0.031), ("Hertfordshire", 0.028), …

---

### Returned Top Passages

*Top-ranked nodes from PPR are highlighted.
**Final answer to this question is highlighted.

**1. Erik Hort**
Erik Hort (born February 16, 1987 in Montebello, New York) is an American soccer player who is currently a Free Agent.

**2. Horton Park (Saint Paul, Minnesota)**
Horton Park is a small arboretum in Saint Paul, Minnesota, United States. Known primarily for its variety of trees, Horton Park has become a symbol of the Saint Paul Midway community.

**3. Montebello, New York**
Montebello (Italian: "Beautiful mountain") is an incorporated village in the town of Ramapo, Rockland County, New York, United States. It is located north of Suffern, east of Hillburn, south of Wesley Hills, and west of Airmont. The population was 4,526 at the 2010 census.

**4. Hertfordshire**
Hertfordshire is the county immediately north of London and is part of the East of England region, a mainly statistical unit. A significant minority of the population across all districts are City of London commuters. To the east is Essex, to the west is Buckinghamshire and to the north are Bedfordshire and Cambridgeshire.

**5. Hull County, Quebec**
Hull County, Quebec is an historic county of Quebec, Canada. It was named after the town of the same name (Hull or Kingston-upon-Hull) in East Yorkshire, England. It is located on the north shore of the Ottawa River and is part of the Outaouais, one of roughly 12 historical regions of Québec.

*Figure 5.* An example of HippoRAG 2 pipeline.

*Table 8.* QA performance (EM / F1 scores) on RAG benchmarks. No retrieval means evaluating the parametric knowledge of the readers. HippoRAG (and HippoRAG 2) uses the denoted LLM for OpenIE (triple filtering) and QA reading.

| Retrieval | Simple QA | | Multi-Hop QA | | | | Discourse Understanding | |
|---|---|---|---|---|---|---|---|---|
| | NQ | PopQA | MuSiQue | 2Wiki | HotpotQA | LV-Eval | NarrativeQA | Avg |
| Llama-3.3-70B-Instruct | | | | | | | | |
| None | 40.2 / 54.9 | 28.2 / 32.5 | 17.6 / 26.1 | 36.5 / 42.8 | 37.0 / 47.3 | 4.0 / 6.0 | 3.4 / 12.9 | 29.7 / 38.4 |
| Contriever (Izacard et al., 2022) | 45.0 / 58.9 | 41.6 / 53.1 | 24.0 / 31.3 | 38.1 / 41.9 | 51.3 / 62.3 | 5.7 / 8.1 | 6.5 / 19.7 | 37.4 / 46.9 |
| BM25 (Robertson & Walker, 1994) | 44.7 / 59.0 | 39.1 / 49.9 | 20.3 / 28.8 | 47.9 / 51.2 | 52.0 / 63.4 | 4.0 / 5.9 | 4.4 / 18.3 | 38.0 / 47.7 |
| GTR (T5-base) (Ni et al., 2022) | 45.5 / 59.9 | 43.2 / 56.2 | 25.8 / 34.6 | 49.2 / 52.8 | 50.6 / 62.8 | 4.8 / 7.1 | 6.8 / 19.9 | 40.0 / 50.4 |
| GTE-Qwen2-7B-Instruct (Li et al., 2023) | 46.6 / 62.0 | 43.5 / 56.3 | 30.6 / 40.9 | 55.1 / 60.0 | 58.6 / 71.0 | 5.7 / 7.1 | 7.9 / 21.3 | 43.8 / 54.9 |
| GritLM-7B (Muennighoff et al., 2025) | 46.8 / 61.3 | 42.8 / 55.8 | 33.6 / 44.8 | 55.8 / 60.6 | 60.7 / 73.3 | 7.3 / 9.8 | 8.2 / 23.9 | 44.9 / 56.1 |
| NV-Embed-v2 (7B) (Lee et al., 2025) | 47.3 / 61.9 | 42.9 / 55.7 | 34.7 / 45.7 | 57.5 / 61.5 | 62.8 / 75.3 | 7.3 / 9.8 | 8.9 / 25.7 | 45.9 / 57.0 |
| RAPTOR (Sarthi et al., 2024) | 36.9 / 50.7 | 43.1 / 56.2 | 20.7 / 28.9 | 47.3 / 52.1 | 56.8 / 69.5 | 2.4 / 5.0 | 5.1 / 21.4 | 38.1 / 48.8 |
| GraphRAG (Edge et al., 2024) | 30.8 / 46.9 | 31.4 / 48.1 | 27.3 / 38.5 | 51.4 / 58.6 | 55.2 / 68.6 | 4.8 / 11.2 | 6.8 / 23.0 | 36.7 / 49.6 |
| LightRAG (Guo et al., 2024) | 8.6 / 16.6 | 2.1 / 2.4 | 0.5 / 1.6 | 9.4 / 11.6 | 2.0 / 2.4 | 0.8 / 1.0 | 1.0 / 3.7 | 4.2 / 6.6 |
| HippoRAG (Gutiérrez et al., 2024) | 43.0 / 55.3 | 42.7 / 55.9 | 26.2 / 35.1 | 65.0 / 71.8 | 52.6 / 63.5 | 6.5 / 8.4 | 4.4 / 16.3 | 42.8 / 53.1 |
| HippoRAG 2 | 48.6 / 63.3 | 42.9 / 56.2 | 37.2 / 48.6 | 65.0 / 71.0 | 62.7 / 75.5 | 9.7 / 12.9 | 8.9 / 25.9 | 48.0 / 59.8 |
| GPT-4o-mini | | | | | | | | |
| None | 35.2 / 52.7 | 16.1 / 22.7 | 11.2 / 22.0 | 30.2 / 36.3 | 28.6 / 41.0 | 3.2 / 5.0 | 2.7 / 14.1 | 22.6 / 33.1 |
| NV-Embed-v2 (7B) (Lee et al., 2025) | 43.5 / 59.9 | 41.7 / 55.8 | 32.8 / 46.0 | 54.4 / 60.8 | 57.3 / 71.0 | 7.3 / 10.0 | 5.1 / 24.2 | 42.9 / 55.7 |
| RAPTOR (Sarthi et al., 2024) | 37.8 / 54.5 | 41.9 / 55.1 | 27.7 / 39.2 | 39.7 / 48.4 | 50.6 / 64.7 | 5.6 / 9.2 | 4.1 / 21.8 | 36.9 / 49.7 |
| GraphRAG (Edge et al., 2024) | 38.0 / 55.5 | 30.7 / 42.0 | 27.0 / 42.0 | 45.7 / 61.0 | 51.4 / 67.6 | 4.9 / 11.0 | 5.4 / 20.9 | 36.0 / 52.6 |
| LightRAG (Guo et al., 2024) | 2.8 / 15.4 | 1.9 / 14.8 | 2.0 / 9.3 | 2.5 / 12.1 | 9.9 / 20.2 | 0.9 / 5.0 | 1.0 / 9.0 | 3.6 / 13.9 |
| HippoRAG (Gutiérrez et al., 2024) | 37.2 / 52.2 | 42.5 / 56.2 | 24.0 / 35.9 | 59.4 / 67.3 | 46.3 / 60.0 | 4.8 / 7.6 | 2.1 / 16.1 | 38.9 / 51.2 |
| HippoRAG 2 | 43.4 / 60.0 | 41.7 / 55.7 | 35.0 / 49.3 | 60.5 / 69.7 | 56.3 / 71.1 | 10.5 / 14.0 | 5.8 / 25.2 | 44.3 / 58.1 |

the triples has failed, where HippoRAG 2 directly uses the results from dense retrieval as a substitute. Overall, though recognition memory is an essential component, the precision of the triple filter has room for further improvement.

**Graph Construction** Graph construction is challenging to evaluate, but we find that only 2% of the samples do not contain any phrases from the supporting passages within the one-hop neighbors of the linked nodes. Given our dense-sparse integration, we can assume that the graphs we construct generally include most of the potentially exploitable information.

**Personalized PageRank** In 50% of the samples, at least half of the linked phrase nodes appear in the supporting documents. However, the final results remain unsatisfactory due to the graph search component. For example, in the second case from Table 11, the recognition memory identifies the key phrase "Philippe, Duke of Orléans" from the query, but the graph search fails to return perfect results among the top-5 retrieved passages.

# F. Cost and Efficiency

For LLM deployment, we run Llama-3.3-70B-Instruct on a machine equipped with four NVIDIA H100 GPUs, utilizing tensor parallelism via vLLM (Kwon et al., 2023).

For a detailed comparison with our baselines, we track the computational resources (# of tokens, indexing time, time per query and GPU memory) usage when indexing and performing QA on the MuSiQue corpus (11k documents) using the Llama-3.3-70B-Instruct model. We compare HippoRAG 2 against NV-Embed-v2 (Lee et al., 2025), RAPTOR (Sarthi et al., 2024), LightRAG (Guo et al., 2024), HippoRAG (Gutiérrez et al., 2024) and GraphRAG (Edge et al., 2024) in Table 12. For the memory requirements, we ignore all memory for model weights since it is shared across all systems.

HippoRAG 2 not only outperforms these RAG methods in QA and retrieval performance but also uses much fewer tokens compared to LightRAG and GraphRAG. In terms of time, we note that HippoRAG 2 is much more efficient than GraphRAG and LightRAG while only being slightly less efficient than both RAPTOR and HippoRAG. HippoRAG 2 's use of fact embeddings does increase its memory requirements compared to our baselines, however, we believe that this is an acceptable tradeoff given our method's performance benefits. Additionally, while all approaches lag behind standard RAG in terms of time and memory efficiency, HippoRAG 2 is the only one that outperforms this strong baseline substantially.

*Table 9.* Passage recall@2 / @5 on RAG benchmarks. * denotes the report from the original paper while we reproduce the HippoRAG results with aligned LLM and retriever.

| | Simple | | Multi-hop | | | |
|---|---|---|---|---|---|---|
| | NQ | PopQA | MuSiQue | 2Wiki | HotpotQA | Avg |
| *Simple Baselines* | | | | | | |
| Contriever (Izacard et al., 2022) | 29.1 / 54.6 | 27.0 / 43.2 | 34.8 / 46.6 | 46.6 / 57.5 | 58.4 / 75.3 | 39.2 / 55.4 |
| BM25 (Robertson & Walker, 1994) | 28.2 / 56.1 | 24.0 / 35.7 | 32.4 / 43.5 | 55.3 / 65.3 | 57.3 / 74.8 | 39.4 / 55.1 |
| GTR (T5-base) (Ni et al., 2022) | 35.0 / 63.4 | 40.1 / 49.4 | 37.4 / 49.1 | 60.2 / 67.9 | 59.3 / 73.9 | 46.4 / 60.7 |
| *Large Embedding Models* | | | | | | |
| GTE-Qwen2-7B-Instruct (Li et al., 2023) | 44.7 / 74.3 | **47.7** / 50.6 | 48.1 / 63.6 | 66.7 / 74.8 | 75.8 / 89.1 | 56.6 / 70.5 |
| GritLM-7B (Muennighoff et al., 2025) | **46.2** / 76.6 | 44.0 / 50.1 | 49.7 / 65.9 | 67.3 / 76.0 | 79.2 / 92.4 | 57.3 / 72.2 |
| NV-Embed-v2 (7B) (Lee et al., 2025) | 45.3 / 75.4 | 45.3 / 51.0 | 52.7 / 69.7 | 67.1 / 76.5 | **84.1** / 94.5 | 58.9 / 73.4 |
| *Structure-augmented RAG* | | | | | | |
| RAPTOR (GPT-4o-mini) | 40.5 / 69.4 | 37.2 / 48.1 | 49.1 / 61.0 | 58.4 / 66.0 | 78.6 / 90.2 | 52.8 / 67.0 |
| RAPTOR (Llama-3.3-70B-Instruct) | 40.3 / 68.3 | 40.2 / 48.7 | 47.0 / 57.8 | 58.3 / 66.2 | 76.8 / 86.9 | 52.5 / 65.6 |
| HippoRAG* (Gutiérrez et al., 2024) | – | – | 40.9 / 51.9 | 70.7 / 89.1 | 60.5 / 77.7 | – |
| HippoRAG (GPT-4o-mini) | 21.6 / 45.1 | 36.5 / 52.2 | 41.8 / 52.4 | 68.4 / 87.0 | 60.1 / 78.5 | 45.7 / 63.0 |
| HippoRAG (Llama-3.3-70B-Instruct) | 21.3 / 44.4 | 40.0 / **53.8** | 41.2 / 53.2 | 71.9 / **90.4** | 60.4 / 77.3 | 47.0 / 63.8 |
| HippoRAG 2 (GPT-4o-mini) | 44.4 / 76.4 | 43.5 / 52.2 | 53.5 / 74.2 | 74.6 / 90.2 | 80.5 / 95.7 | 59.3 / 77.7 |
| HippoRAG 2 (Llama-3.3-70B-Instruct) | 45.6 / **78.0** | 43.9 / 51.7 | **56.1** / **74.7** | **76.2** / **90.4** | 83.5 / **96.3** | **61.1** / **78.2** |

# G. Implementation Details and Hyperparameters

## G.1. HippoRAG 2

We provide a detailed explanation of the PPR initialization process used in HippoRAG 2 here. The key goal is to determine the seed nodes for the PPR search and assign appropriate reset probabilities to ensure an effective retrieval process.

**Seed Node Selection** The seed nodes for the PPR search are categorized into two types: phrase nodes and passage nodes. All the scores given by the embedding model below use normalized embedding to calculate. 1) Phrase Nodes: These seed nodes are selected from the phrase nodes within the filtered triples, which are obtained through the recognition memory component. If recognition memory gives an empty triple list and no phrase node is available, HippoRAG 2 directly returns top passages using the embedding model without any graph search. Otherwise, we keep at most 5 phrase nodes as the seed nodes, and the ranking score of each phrase node is computed as the average score of all filtered triples it appears in. 2) Passage Nodes: Each passage node is initially scored using an embedding-based similarity, and these scores are processed as follows. All passage nodes are taken as seed nodes since we find that activating a broader set of potential passages is more effective for uncovering passages along multi-hop reasoning chains compared to focusing only on the top-ranked passages.

**Reset Probability Assignment** After determining the seed nodes, we assign reset probabilities to control how likely the PPR algorithm will return to these nodes during the random walk. The rules are: 1) Phrase nodes receive reset probabilities directly as their ranking scores. 2) Passage nodes receive reset probabilities proportional to their embedding similarity scores, i.e., to balance the influence of phrase nodes and passage nodes, we apply a weight factor to the passage node scores. Specifically, the passage node scores are multiplied by the weight factor discussed in Section 6.2. This ensures that passage nodes and phrase nodes contribute appropriately to the retrieval process.

**PPR Execution and Passage Ranking** Once the seed nodes and their reset probabilities are initialized, we run PPR over the constructed graph. The final ranking of passages is determined based on the PageRank scores of the passage nodes. Top-ranked passages are then used as inputs for the downstream QA reading process. We manage our KG and run the PPR algorithm using the python-igraph library.[2]

By incorporating both phrase nodes and passage nodes into the PPR initialization, our approach ensures a more effective retrieval of relevant passages, especially for multi-hop reasoning tasks.

---

[2] https://python.igraph.org/en/stable/

*Table 10.* Knowledge graph statistics using different LLMs for OpenIE. The nodes and triples are counted based on unique values.

| | NQ | PopQA | MuSiQue | 2Wiki | HotpotQA | LV-Eval | NarrativeQA |
|---|---|---|---|---|---|---|---|
| Llama-3.3-70B-Instruct | | | | | | | |
| # of phrase nodes | 68,375 | 76,539 | 85,288 | 44,004 | 81,200 | 175,195 | 9,224 |
| # of passage nodes | 9,633 | 8,676 | 11,656 | 6,119 | 9,811 | 22,849 | 4,111 |
| # of total nodes | 78,008 | 85,215 | 96,944 | 50,123 | 91,011 | 198,044 | 13,335 |
| # of extracted edges | 125,777 | 124,579 | 140,830 | 68,881 | 130,058 | 314,324 | 26,208 |
| # of synonym edges | 899,031 | 845,014 | 1,125,951 | 593,298 | 994,187 | 2,674,833 | 72,494 |
| # of context edges | 126,757 | 118,909 | 132,586 | 64,132 | 122,437 | 375,424 | 33,395 |
| # of total edges | 1,151,565 | 1,088,502 | 1,399,367 | 726,311 | 1,246,682 | 3,364,581 | 132,097 |
| GPT-4o-mini | | | | | | | |
| # of phrase nodes | 86,904 | 85,744 | 101,641 | 49,544 | 95,105 | 217,085 | 15,365 |
| # of passage nodes | 9,633 | 8,676 | 11,656 | 6,119 | 9,811 | 22,849 | 4,111 |
| # of total nodes | 96,537 | 94,420 | 113,297 | 55,663 | 104,916 | 239,934 | 19,476 |
| # of extracted edges | 114,900 | 108,989 | 125,903 | 62,626 | 119,630 | 303,491 | 24,373 |
| # of synonym edges | 1,094,651 | 901,528 | 1,304,605 | 715,763 | 1,126,501 | 3,268,084 | 14,075 |
| # of context edges | 142,419 | 127,568 | 146,293 | 68,348 | 133,220 | 404,210 | 38,632 |
| # of total edges | 1,351,970 | 494,082 | 1,576,801 | 846,737 | 1,379,351 | 3,975,785 | 77,080 |

*Table 11.* Two examples from MuSiQue where passage recall@5 is less than 1.0.

| | |
|---|---|
| **Query** | Where is the district that the person who wanted to reform and address Bernhard Lichtenberg's religion preached a sermon on Marian devotion before his death located? |
| **Answer** | Saxony-Anhalt |
| **Supporting Passages (Title)** **Retrieved Passages (Title)** | 1. Mary, mother of Jesus 2. Reformation 3. Wittenberg (district) 4. Bernhard Lichtenberg **1. Bernhard Lichtenberg 2. Mary, mother of Jesus** 3. Ambroise-Marie Carré **4. Reformation** 5. Henry Scott Holland (Recall@5 is 0.75) |
| **Query to Triple (Top-5)** | ("Bernhard Lichtenberg", "was", "Roman Catholic Priest") ("Bernhard Lichtenberg", "beatified by", "Catholic Church") ("Bernhard Lichtenberg", "died on", "5 November 1943") ("Catholic Church", "beatified", "Bernhard Lichtenberg") ("Bernhard Lichtenberg", "was", "Theologian") All above subjects and objects appear in supporting passages |
| **Filtered Triple** | Empty |
| **Query** | Who is the grandmother of Philippe, Duke of Orléans? |
| **Answer** | Marie de' Medici |
| **Supporting Passages (Title)** **Retrieved Passages (Title)** | 1. Philippe I, Duke of Orléans 2. Leonora Dori **1. Philippe I, Duke of Orléans** 2. Louise Élisabeth d'Orléans 3. Philip III of Spain 4. Anna of Lorraine 5. Louis Philippe I (Recall@5 is 0.5) |
| **Query to Triple (Top-5)** | ("Bank of America", "purchased", "Fleetboston Financial") ("Fleetboston Financial", "was acquired by", "Bank of America") ("Bank of America", "acquired", "Fleetboston Financial") ("Bank of America", "announced purchase of", "Fleetboston Financial") ("Bank of America", "merged with", "Fleetboston Financial") All above subjects and objects appear in supporting passages |
| **Filtered Triple** | ("Bank of America", "purchased", "Fleetboston Financial") ("Fleetboston Financial", "was acquired by", "Bank of America") All above subjects and objects appear in supporting passages |

*Table 12.* We report the computational resource requirements (indexing tokens, indexing time, time per query, GPU memory requirements during QA) for the RAG baselines on the MuSiQue corpus (11,656 passages). For each metric, we include the percentage that this method obtains with respect to the HippoRAG 2 metric (100%).

|  | NV-Embed-v2 | RAPTOR | LightRAG | GraphRAG | HippoRAG | HippoRAG 2 |
|---|---|---|---|---|---|---|
| **Input Tokens** | – | 1.7M (18.5%) | 68.5M (744.6%) | 115.5M (1255.4%) | 9.2M (100.0%) | 9.2M (100.0%) |
| **Output Tokens** | – | 0.2M (6.7%) | 18.3M (610.0%) | 36.1M (1203.3%) | 3.0M (100.0%) | 3.0M (100.0%) |
| **Indexing Time (min)** | 12.1 (12.3%) | 100.5 (101.0%) | 235.0 (236.2%) | 277.0 (278.4%) | 57.5 (57.7%) | 99.5 (100.0%) |
| **QA Time/Query (sec)** | 0.3 (25.0%) | 0.6 (50.0%) | 13.3 (1008.3%) | 10.7 (891.7%) | 0.9 (75.0%) | 1.2 (100.0%) |
| **QA GPU Memory (GB)** | 1.7 (17.2%) | 1.4 (14.1%) | 4.5 (45.5%) | 3.7 (37.4%) | 6.0 (60.6%) | 9.9 (100.0%) |

**Hyperparameters** We perform hyperparameter tuning on 100 examples from MuSiQue's training data. The hyperparameters are listed in Table 13.

*Table 13.* Hyperparameters set on HippoRAG 2

| Hyperparameter | Value |
|---|---|
| Synonym Threshold | 0.8 |
| Damping Factor of PPR | 0.5 |
| Temperature | 0.0 |

## G.2. Comparison Methods

We use PyTorch (Paszke et al., 2019) and HuggingFace (Wolf et al., 2019) for dense retrievers and BM25s (Lù, 2024) for the BM25 implementation. For GraphRAG (Edge et al., 2024) and LightRAG (Guo et al., 2024), we adhere to their default hyperparameters and prompts. To ensure a consistent evaluation, the same QA prompt that HippoRAG 2 adopts from HippoRAG (Gutiérrez et al., 2024) is applied to rephrase the original response of GraphRAG and LightRAG.

**Hyperparameters** We keep the default indexing hyperparameters for GraphRAG and LightRAG. For QA, we perform hyperparameter tuning on the same 100 samples as Appendix G.1.

*Table 14.* Hyperparameters set on GraphRAG and LightRAG

| Hyperparameters | GraphRAG | LightRAG |
|---|---|---|
| Mode | Local | Local |
| Response Type | Short phrase | Short phrase |
| Top-k Phrases for QA | 60 | 60 |
| Chunk Token Size | 1,200 | 1,200 |
| Chunk Overlap Token Size | 100 | 100 |
| Community Report Max Length | 2,000 | – |
| Max Input Length | 8,000 | – |
| Max Cluster Size | 10 | – |
| Entity Summary Max Tokens | – | 500 |

