# OpenReview forum: "From RAG to Memory: Non-Parametric Continual Learning for Large Language Models"
_ICML.cc/2025/Conference — ICML 2025 poster_

### Official Review · Reviewer_5bA8 · 2025-03-05

**Overall Recommendation:** 3

**Summary:**

The paper introduces *ContinualRAG*, a novel retrieval-augmented generation (RAG) framework designed to enhance large language models (LLMs) with a human-like long-term memory system for non-parametric continual learning. Building on the HippoRAG framework, ContinualRAG aims to address limitations in standard RAG by improving performance across three key memory dimensions: *factual memory* (simple QA tasks), *sense-making* (interpreting complex contexts), and *associativity* (multi-hop reasoning). Its main algorithmic contributions include:
1. Dense-Sparse Integration: Incorporating passage nodes into the knowledge graph (KG) alongside phrase nodes to better capture context, inspired by dense and sparse coding in human memory.
2. Deeper Contextualization: Using a query-to-triple matching approach instead of entity-centric methods to improve query-KG alignment.
3. Recognition Memory: Adding an LLM-based triple filtering step to refine retrieved triples for graph search.

The authors evaluate ContinualRAG on benchmarks like NaturalQuestions (NQ), PopQA, MuSiQue, 2WikiMultiHopQA, HotpotQA, LV-Eval, and NarrativeQA.

**Claims And Evidence:**

The paper’s primary claims are well-supported by empirical evidence, though some areas could benefit from deeper analysis:
- **Claim: ContinualRAG outperforms standard RAG across all memory tasks.**
  - **Evidence**: Tables 2 and 3 provide F1 scores and recall@5 metrics across seven benchmarks, showing consistent superiority over baselines (e.g., 63.3 vs. 61.9 on NQ, 48.6 vs. 45.7 on MuSiQue). Ablation studies (Table 4) further validate the contributions of each component (e.g., query-to-triple improves recall@5 by 12.5% over NER-to-node).
  - **Weakness**: The claim of "comprehensive" outperformance lacks statistical significance tests to confirm robustness across runs or datasets.

Overall, the evidence is convincing, but transparency in metric aggregation and statistical validation could strengthen the claims.

**Essential References Not Discussed:**

No

**Experimental Designs Or Analyses:**

- **QA Performance (Table 2)**: Sound design comparing ContinualRAG against diverse baselines (BM25, NV-Embed-v2, HippoRAG) using Llama-3.3-70B-Instruct. The use of F1 scores across seven datasets is valid, but the lack of error bars or multiple runs limits reliability assessment.
- **Retrieval Performance (Table 3)**: Recall@5 is a standard metric, and reproducing HippoRAG with the same setup ensures fairness. However, the original HippoRAG paper reports higher R@5 (89.1 vs. 79.4 on 2Wiki), suggesting a potential issue in reproduction fidelity.
- **Ablation Study (Table 4)**: Well-designed to isolate contributions (e.g., query-to-triple vs. NER-to-node), with clear recall@5 improvements. The omission of filtering in NER-to-node baselines is justified but could be explored further.

**Issue**: No statistical significance tests are reported, weakening confidence in the results’ robustness.

**Methods And Evaluation Criteria:**

Yes

**Other Comments Or Suggestions:**

No.

**Other Strengths And Weaknesses:**

- **Strengths**:
  - Integration of passage nodes and recognition memory builds on HippoRAG.
  - Well-structured, with figures (e.g., Figure 2) and tables enhancing readability.
  - Broad applicability across memory types.

- **Weaknesses**:
  - Heavy reliance on HippoRAG reduces novelty; In my opinion the only major difference with the former is the passage integration part.

**Questions For Authors:**

1. Why were no statistical tests reported? Adding them could confirm result reliability—would significant p-values change your confidence in ContinualRAG’s superiority?

**Relation To Broader Scientific Literature:**

ContinualRAG builds on:
- *HippoRAG*: Extends its PPR and OpenIE approach, adding passage integration and recognition memory to address context loss.
- *RAG Evolution*: Cites standard RAG (Zhong et al., 2023) and structure-augmented methods (RAPTOR, GraphRAG), positioning itself as a comprehensive solution.
- *Neurobiology*: Links to hippocampal indexing (Klein et al., 2006; Suzuki, 2005) mirror HippoRAG’s inspiration, grounding it in memory theory.

**Theoretical Claims:**

The paper does not present formal proofs but draws theoretical inspiration from neurobiology (e.g., dense-sparse coding, recognition memory). It remains conceptual rather than rigorously proven.

---

> ### Author Rebuttal · Authors · 2025-03-31
>
> We are very thankful for the reviewer’s kind acknowledgment of our work as convincing, well-structured and broadly applicable across memory types. We will address the reviewer’s careful suggestions in the sections below.
>
> ## Statistical Significance Testing
>
> > Why were no statistical tests reported? Adding them could confirm result reliability—would significant p-values change your confidence in ContinualRAG’s superiority?
>
> We thank the reviewer for pointing out the need for statistical significance testing to ensure the reliability of our results. In order to ascertain the significance of ContinualRAG’s performance over NV-Embed-V2 (the best performing baseline), we run a simple bootstrapping statistical test. More specifically, we created 10,000 different datasets by sampling from each set of answers with replacement and thus obtained a distribution over the differences in their QA performance.
>
> Through this method, we find that ContinualRAG’s performance is significantly larger than that of NV-Embed-V2 (p-value < 0.05) in 4 out of 7 datasets. Additionally, NV-Embed-V2 does not significantly outperform ContinualRAG in the other 3 datasets, demonstrating that ContinualRAG is robustly superior to our strongest baseline. We will include these significance testing results in the camera-ready version of our paper.
>
> | Dataset           | p-value  |
> |-------------------|----------|
> | 2WikiMultiHopQA  | 0.0000   |
> | MuSiQue          | 0.0013   |
> | NQ          | 0.0209   |
> | LVEval           | 0.0484   |
>
> ## Novelty Concerns
>
> > Heavy reliance on HippoRAG reduces novelty; In my opinion the only major difference with the former is the passage integration part.
>
> We appreciate the reviewer’s perspective, however, we would like to clarify our view on our work’s originality. While it is true that we build upon the existing HippoRAG framework, our approach systematically explores and enhances key components of the system in ways that are both principled and non-trivial. The space of possible design choices in structure-augmented RAG systems is vast, and identifying which modules to modify and how to do so effectively is itself a meaningful research contribution.
>
> Our modifications are not superficial; rather, they are informed by clear hypotheses about the limitations of the original modules and are validated through rigorous ablation studies (Table 4), which demonstrate substantial and consistent improvements in performance. We believe that such targeted, evidence-based improvements can be as impactful as proposing entirely new frameworks, especially when they advance the capabilities of an already influential baseline.
>
> Moreover, our work provides actionable insights into the design of RAG systems that others in the community can build upon—offering a path forward for both incremental and architectural innovations.
>
> ## Reproduction Fidelity
>
> > Retrieval Performance (Table 3): Recall@5 is a standard metric, and reproducing HippoRAG with the same setup ensures fairness. However, the original HippoRAG paper reports higher R@5 (89.1 vs. 79.4 on 2Wiki), suggesting a potential issue in reproduction fidelity.
>
> We would like to note that our reproduced HippoRAG results, reported on Table 3, are very close (or slightly better) than the ones reported in the original paper due to the use of a stronger embedding model and LLM for OpenIE. We changed both models in order to effectively compare with all other baselines.
>
> More specifically, the R@5 scores reported in the original HippoRAG paper are 51.9, 89.1 and 77.7 respectively for MuSiQue, 2Wiki and HotpotQA. Our reproduced HippoRAG R@5 scores are 53.2, 90.4 and 77.3 for the same three datasets.
>
> We ask the reviewer to let us know if they have any other questions concerning our reproduced HippoRAG results.
>
> ## Filtering & Query-to-Node
>
> > Ablation Study (Table 4): Well-designed to isolate contributions (e.g., query-to-triple vs. NER-to-node), with clear recall@5 improvements. The omission of filtering in NER-to-node baselines is justified but could be explored further.
>
> Although the reviewer makes an interesting point there are a few reasons why we omit a filter for NER-to-node:
> - NER-to-node is already using an LLM to extract named entities before retrieving nodes.
> - Though a post-extraction filter could be added to NER-to-node, it would require designing a different filtering methodology specifically for this module.
> - Given that the performance of ContinualRAG w/o filter (avg. 86.4) is already much better than with NER-to-node (avg. 74.6), as shown in Table 4, we believe that designing such a filter was not the most worthwhile direction to explore.

---

### Official Review · Reviewer_jXKH · 2025-03-08

**Overall Recommendation:** 3

**Summary:**

This paper proposes ContinualRAG that improves the performance of RAG on natural question answering and multi-hop reasoning benchmarks.

The method builds on the prior work, HippoRAG, which performs reasoning on a knowledge graph constructed at an offline phase. In the offline indexing phase, HippoRAG extracts knowledge triples of named-entities and detects synonyms to create additional edges in the knowledge graph. In the online retrieval phase, a query is deconstructed to its named-entities, the knowledge graph is queried for the named entities and their synonyms, then Personalized PageRank is used to retrieve the information from the graph as the response to the query.

This work argues HippoRAG is limited because it is entity-centric and loses information both during indexing and inference. As such, ContinualRAG makes the following modifications to HippoRAG:
1) Adds passage nodes to the knowledge graph that connect to named-entity nodes in the passage with context edges. The goal is to have more contextual information in the KG.
2) Improves the linking of queries to the KG from HippoRAG’s named-entity-recognition to a Query-to-triple approach which matches the entire query to triples in the graph using text embeddings.
3) Improve the retrieval step to two 1) retrieve top-k triples from the graph using an embedding model and 2) filter triples using an LLM.

Given these improvements, for QA evaluations the online retrieval of ContinualRAG involves assigning scores to retrieved passage and seed nodes and then executing Personalized PageRank (PPR) to retrieve the top-ranked passages as the answer to the question.

The evaluations are divided into QA and retrieval evaluations. On QA evaluations, they demonstrate on average 6.7% improvement on HippoRAG. Their improvement brings them above the performance of large embedding models utilizing 7B LLMs by 2.8%. On retrieval evaluations they achieve 15% improvement on HippoRAG and nearly 5% better than large embedding models.

The ablations in Table 4 demonstrate the importance of all 3 proposed modifications. Particularly, the query-triple approach for linking the queries to the KG accounts for the majority of the improvement on most evaluations (15% on average) except for 2Wiki. The other two modifications of adding passage nodes (6% on average) and LLM filtering in retrieval (0.7% on average) are also effective.

## Update after rebuttal
I thank authors for their response and recommend incorporating the clarifications into a revision. I maintain my rating of weak accept conditioned on applying needed improvements to the text to improve clarity.

**Claims And Evidence:**

The paper claims their 3 modifications to HippoRAG improve the contextual-awareness of the method. Substantial improvements across a diverse set of evaluations, along with ablation studies, substantiate their claim.

**Essential References Not Discussed:**

If the paper is position itself as continual method as the title may suggest, it should consider expanding the related works section to discuss more continual pretraining methods.
- Roth, Karsten, et al. "A Practitioner's Guide to Continual Multimodal Pretraining." arXiv preprint arXiv:2408.14471 (2024).
- Li, Jeffrey, et al. "Tic-lm: A multi-year benchmark for continual pretraining of language models."

**Experimental Designs Or Analyses:**

Lines 370-373 state that the method utilizes Llama-3.3-70B Instruct for extraction and triple filtering. However, the tables refer to models that utilize 7B parameter LLMs as large embedding models. The relation to such methods and why they are referred to as large embedding models while the method utilizes 70B models albeit at a limited capacity is not clear.

It is also not clear what the impact of the model size is in the proposed method. Table 9 in the appendix provides some ablations with GPT-4o-mini but does not ablate on different model sizes of the Llama-3.3 family. Prior work HippoRAG provided some ablations between Llama-3.1 8B/70B models but the importance of these models in ContinualRAG might be different.

**Methods And Evaluation Criteria:**

The proposed method involves three well-justified modifications to HippoRAG. The evaluations are similar to prior work and evaluate the reasoning and question-answering capability of the model.

**Other Comments Or Suggestions:**

- Line 355: N otably ->Notably
- Line 427: It combining -> It combines

**Other Strengths And Weaknesses:**

- The paper is sparse in details about prior works such as HippoRAG. In particular, understanding section 3.1 is crucial for the understanding of the contributions, however, definitions and details on the following terms are missing: OpenIE, Personalized PageRank, PHR, reset probability, etc.
- The paper is missing descriptions for baselines in section 4.1. At least a description of groups of methods and a comparison between the bottlenecks and capacities of methods to the proposed method is required.
- A clear description of the evaluations and execution difference between QA and retrieval setups is missing.
- The paper is named “ContinualRAG”, however, the method does not seem to have any “continual” aspect.

**Questions For Authors:**

N/A

**Relation To Broader Scientific Literature:**

This paper advances the question-answering capabilities that require a knowledge graph to answer factual questions.

**Theoretical Claims:**

N/A. The paper makes no theoretical claims.

---

> ### Author Rebuttal · Authors · 2025-04-01
>
> We would like to thank the reviewer for thoroughly reviewing our work and noting that our modifications are well-justified and bring strong improvements over all our baselines.
>
> ## ContinualRAG is a Continual Learning Method
>
> > The paper is named “ContinualRAG”, however, the method does not seem to have any “continual” aspect.
>
> As we argued in our work, RAG has become the de facto solution for continual learning in LLMs due to its simplicity and strong performance when compared to parametric alternatives. Our methodology builds on these already robust continual learning systems, enabling them to handle tasks that require more in-depth learning of new knowledge, such as associativity and sense-making. Given its strong performance on these demanding tasks, we argue that ContinualRAG system not only classifies as a continual learning system but also elevates the standard of what it means to continually learn.
>
> To further show this, we provide an experiment that directly evaluates our method’s performance as more knowledge is aggregated. We refer the reviewer to the *Continual Knowledge Injection Experiments* section in our response to **reviewer gi53** above for experimental details, results and discussion.
>
> For a more detailed explanation of motivation being our experiments, we refer the reviewer to the *Non-Parametric Continual Learning for LLMs: Ideal Experiments* section of our response to **reviewer K3hz**.
>
> ## Why are 7B embedding models “large”?
>
> > The method utilizes Llama-3.3-70B Instruct for extraction and triple filtering. However, the tables refer to models that utilize 7B parameter LLMs as large embedding models. The relation to such methods and why they are referred to as large embedding models while the method utilizes 70B models albeit at a limited capacity is not clear.
>
> We refer to 7B embedding models as ''large'' because they are much larger than classic ones like GTR (335M) and Contriever (110M). Although smaller than LLMs, they are the largest and strongest models available on the MTEB benchmark.
>
> ## How does LLM size impact ContinualRAG?
>
> > It is also not clear what the impact of the model size is in the proposed method. Table 9 in the appendix provides some ablations with GPT-4o-mini but does not ablate on different model sizes of the Llama-3.3 family.
>
> We report results using Llama-3.1-8B for ContinualRAG (Llama 3.3 family only has 70B models). It shows the 8B model is not sufficiently capable of supporting our system on both types of tasks (MuSiQue & NQ).
>
> | Model                               | MuSiQue |   NQ   |
> |---|---|---|
> | NV-Embed-v2 (7B)                   |  45.70  | 61.90  |
> | ContinualRAG (Llama-3.1-8B-Instruct) |  37.93  | 55.28  |
> | ContinualRAG (Llama-3.3-70B-Instruct) |  48.60  | 63.30  |
>
> ## Clarifications
>
> > The paper is sparse in details about prior works such as HippoRAG. In particular, understanding section 3.1 is crucial for the understanding of the contributions, however, definitions and details on the following terms are missing: OpenIE, Personalized PageRank, PHR, reset probability, etc.
>
> **OpenIE:** OpenIE extracts entity–relation–entity triples from text without predefined relation types, in contrast to standard IE.
>
> **Personalized PageRank (PPR):** PPR is a variation of PageRank that measures the importance of nodes in a graph relative to a source node (or source nodes).
>
> **Reset Probabilities:** This vector quantifies the importance of source nodes for PPR.
>
> **Parahippocampal regions (PHR):** This terminology is borrowed from the HippoRAG paper, which is an analogy between their retrieval encoder and the parahippocampal regions of the human brain.
>
> > The paper is missing descriptions for baselines in section 4.1. At least a description of groups of methods and a comparison is required.
>
> We include several RAG baselines like BM25, popular retrievers (Contriever, GTR) and 3 SoTA models from MTEB. RAPTOR, GraphRAG, and LightRAG use summarization to enable sense-making capacity. HippoRAG performs well on associative tasks.
>
> > A clear description of the evaluations and execution difference between QA and retrieval setups is missing.
>
> Our QA module uses the top-5 retrieved passages as context for an LLM (GPT-4o-mini or Llama-3.3-70B-Instruct) to generate the final answer. The QA result is evaluated by token-based EM/F1 scores, aligning with MuSiQue/2Wiki/HotpotQA official metrics.
>
> ## Continual Pretraining Related Work
>
> > If the paper is position itself as continual method as the title may suggest, it should consider expanding the related works section to discuss more continual pretraining methods.
> > - Roth, Karsten, et al. "A Practitioner's Guide to Continual Multimodal Pretraining." arXiv preprint arXiv:2408.14471 (2024).
> > - Li, Jeffrey, et al. "Tic-lm: A multi-year benchmark for continual pretraining of language models."
>
> We thank the reviewer for highlighting these important works, we will incorporate them into Section 2.1 of our camera-ready version.

---

> > ### Comment · Reviewer_jXKH · 2025-04-03
> >
> > I thank authors for their response and recommend incorporating the clarifications into a revision. I maintain my rating of weak accept conditioned on applying needed improvements to the text to improve clarity.

---

### Official Review · Reviewer_K3hz · 2025-03-13

**Overall Recommendation:** 2

**Summary:**

This paper presents a method to enhance traditional RAG models for large language models. The proposed approach is based on HippoRAG and introduces a combination of phrase nodes and passage nodes, inspired by how human memory represents and processes information at different granularities. Additionally, the method incorporates query-to-triple contextualization, tightly associating user queries with knowledge graph nodes. Experimental results demonstrate that the proposed approach achieves a 7% performance gain over state-of-the-art embedding models in associative memory while maintaining superior factual recall and discourse understanding.

**Claims And Evidence:**

The dense-sparse integration, inspired by human memory processing at different granularities, is well-motivated. However, it seems questionable to connect all edges to phrases linked to passages. While this approach may work well for small documents, it poses a scalability issue, when dealing with numerous documents, the KG can grow excessively large. Moreover, the method does not account for temporal changes in knowledge, which could limit its practical applicability.

**Essential References Not Discussed:**

- Most of the relevant studies are appropriately cited.

**Experimental Designs Or Analyses:**

- If the proposed method is intended for RAG-based question answering systems, the experimental design and ablation studies appear valid and sound.

- However, since this approach stores more information in the KG compared to the existing structure-augmented RAG, an analysis is required to determine how much additional storage and computational resources are needed.

**Methods And Evaluation Criteria:**

- The evaluation methods and datasets used for comparison with existing RAG systems are meaningful. However, they do not adequately address experiments related to continual learning.

- If the proposed method is intended for continual learning, it is crucial to evaluate how the knowledge graph updates over time such that new documents are continuously introduced. This includes its impact on offline indexing, search, and question-answering.

- Additionally, the proposed method modifies only certain modules of the existing HippoRAG framework. As a result, it does not qualify as a novel framework, as suggested in the conclusion.

**Other Comments Or Suggestions:**

- If this method is designed for continual learning, the experimental setup should be revised to incorporate a temporal dimension.

- If the paper focuses on traditional RAG systems, the study should emphasize document storage, retrieval, and utilization rather than continual learning.

**Other Strengths And Weaknesses:**

- Strength: The study effectively integrates cognitive-inspired memory representation into the RAG model, reflecting how the brain processes information at different levels of granularity.

- Weakness: The originality of this work is somewhat limited, as it primarily modifies existing modules in HippoRAG rather than introducing an entirely new framework.

**Questions For Authors:**

- How much additional time and computational resources does this approach require compared to the existing models used for comparison?

**Relation To Broader Scientific Literature:**

- The proposed method improves entity-centric retrieval in HippoRAG and enhances alignment in knowledge graphs. Experimental results indicate that the QA performance improves by approximately 3 percentage points compared to SOTA models.

**Theoretical Claims:**

There are no theoretical claims in this paper.

---

> ### Author Rebuttal · Authors · 2025-04-01
>
> We sincerely appreciate the reviewer’s detailed comments and questions, they will surely enhance the quality of our work. We are happy to know that the reviewer found our human memory inspired methodology well-motivated and our experimental settings meaningful, valid and sound.
>
> ## Non-Parametric Continual Learning for LLMs: Ideal Experiments
>
> > If this method is designed for continual learning, the experimental setup should be revised to incorporate a temporal dimension.
>
> We appreciate the reviewer’s suggestion to add a temporal dimension to our experiments; however, we believe that our current setup appropriately evaluates the continual learning abilities of non-parametric methods given their unique strengths and limitations.
>
> As explained in our paper, non-parametric methods have become the de facto continual learning solution for LLM due to their simplicity and strong performance. In standard continual learning benchmarks, which measure catastrophic forgetting and simple factual learning, standard RAG outperforms parametric alternatives like model editing and continual pretraining by substantial margins (MQuAKE, EvolvingQA).
>
> In contrast to their strong performance in these simpler settings, non-parametric methods struggle with richer forms of continual learning. More specifically, given that standard RAG acquires isolated knowledge, it is limited in its capacity to enable complex tasks over new knowledge, such as associativity and sense-making. Our experimental setup, which consists of a set of such tasks, is thus designed to explore this limitation in non-parametric continual learning methods.
>
> That said, we agree that assessing performance as more knowledge is incrementally introduced would be a valuable addition to our paper. We refer the reviewer to the *Continual Knowledge Injection Experiments* section in our response to **reviewer gi53** above for experimental details, results and discussion.
>
> We will add this discussion and experiment to the camera-ready version.
>
> ## ContinualRAG’s Computational Overhead
>
> > How much additional time and computational resources does this approach require compared to the existing models used for comparison?
>
> We appreciate the reviewer’s question concerning ContinualRAG’s efficiency compared to our baselines. To address this, we report the time and memory resources required for offline indexing and online retrieval, we will add them to Appendix F alongside the token-level costs reported in Table 12. For indexing, we indexed 11k documents using a Llama-3.3-70B model using vLLM on 4 H100s. For the memory requirements, we ignore all memory for model weights since it is shared across all systems.
>
> | Model        | Indexing Time (min) | QA Time per Query (s) | QA Memory (GB) |
> |---|---|---|---|
> | NV-Embed-V2  | 12.12               | 0.33                  | 1.7            |
> | RAPTOR       | 100.50              | 0.61                  | 1.4            |
> | GraphRAG     | 276.95              | 10.70                 | 3.7            |
> | LightRAG     | 234.95              | 13.31                 | 4.5            |
> | HippoRAG     | 57.50               | 0.90                  | 6.0            |
> | ContinualRAG | 99.50               | 1.15                  | 9.9            |
>
>
> As we can see, in terms of time, ContinualRAG is much more efficient than GraphRAG and LightRAG and only slightly less efficient than both RAPTOR and HippoRAG. For memory usage, ContinualRAG’s use of fact embeddings does increase its requirements, however, we believe this is acceptable given its performance benefits. Additionally, while all approaches lag behind standard RAG in terms of time and memory efficiency, ContinualRAG is the only one that outperforms this strong baseline substantially.
>
> ## Novelty Concerns
>
> > The originality of this work is somewhat limited, as it primarily modifies existing modules in HippoRAG rather than introducing an entirely new framework.
>
> We appreciate the reviewer’s perspective, however, we would like to clarify our view on our work’s originality. While it is true that we build upon the existing HippoRAG framework, our approach systematically explores and enhances key components of the system in ways that are both principled and non-trivial. The space of possible design choices in structure-augmented RAG systems is vast, and identifying which modules to modify and how to do so effectively is itself a meaningful research contribution.
>
> Our modifications are not superficial; rather, they are informed by clear hypotheses about the limitations of the original modules and are validated through rigorous ablation studies (Table 4), which demonstrate substantial and consistent improvements in performance. We believe that such targeted, evidence-based improvements can be as impactful as proposing entirely new frameworks, especially when they advance the capabilities of an already influential baseline.

---

> > ### Comment · Reviewer_K3hz · 2025-04-04
> >
> > I appreciate the authors' response and recommend incorporating the clarifications into a revised version. Thank you, in particular, for providing the results of additional experiments on knowledge injection within a short time frame. However, I am curious whether one of the experiments follows the same setting as the existing continual learning-based QA, specifically dividing the full corpus into four parts.
> >
> > That said, I disagree with the authors' statement that ``non-parametric methods have become the de facto continual learning solution for LLMs.'' The authors appear to formulate the task based on the assumption that LLMs cannot be updated. However, essential continual learning for LLMs is fundamentally different, and this distinction needs to be clarified.
> >
> > Furthermore, I still perceive the added module as primarily an incremental novelty rather than a fundamentally new combination of structure-aware and dense-based RAG systems (such as a combined method based on HippoRAG). Therefore, I find it difficult to agree that the research contribution is as meaningful as claimed by the authors.
> >
> > For these reasons, I maintain my original rating.

---

> > > ### Author Response · Authors · 2025-04-08
> > >
> > > We sincerely appreciate the reviewer’s thoughtful comment, as it allows us to further clarify the conceptual motivation behind our work.
> > >
> > > ## The Role of Continual Learning in LLMs
> > >
> > > Continual learning has long been a foundational pursuit of AI research, aiming to allow machine learning models to **learn from new data** without **forgetting** what they previously learned. In recent years, LLMs have been shown to attain remarkable zero-shot capabilities across a wide variety of tasks and domains simultaneously. This impressive adaptability has made traditional domain or task-specific continual learning methods less relevant for LLMs, however, their **inability to continuously absorb new factual knowledge** has opened a crucial new line of research in continual learning within these models.
> > >
> > > Many parametric continual learning methods, such as **model editing** and **continual training**, have thus been developed to update factual knowledge within LLMs. However, these methods have faced **severe practical constraints**. First, while model editing appears to be a promising solution for updating a small number of facts, these updates are not reflected outside of a narrow set of contexts (Zhong et al. 2023, Wang et al. 2024) and lead to catastrophic forgetting as the process is repeated for more facts (Gupta et al. 2024). Meanwhile, continual training of LLMs suffers from similar ineffectiveness while being prohibitively expensive as well.
> > >
> > > Given these challenges, **retrieval-augmented generation (RAG)** has emerged as an effective and efficient solution for **continual learning in LLMs**. RAG sidesteps concerns of parametric updates, allowing systems to add new facts **without risking catastrophic forgetting** by changing the LLM. Moreover, most deployed LLMs (e.g., ChatGPT, Perplexity) retrieve web knowledge to support generation, reinforcing non-parametric methods as the de facto continual learning solution in practice.
> > >
> > > In summary, our claim is not that parametric continual learning lacks merit or is impossible—but rather that in the context of LLMs, RAG has become the most viable and widely adopted means of maintaining LLM systems factually up-to-date.
> > >
> > > ## Continual Learning in LLMs Beyond Factual Recall
> > >
> > > Nevertheless, while *RAG* excels at simple factual tasks, it has **major limitations** in more complex continual learning abilities—such as **associative reasoning** and **sense-making**. This is the gap our work addresses. Our focus is not to further demonstrate that RAG can continuously integrate new factual knowledge but to provide a non-parametric solution that addresses these deficiencies in LLM continual learning. Thus, our experimental setting is designed to evaluate each method’s ability to **use new knowledge in associative, discourse-level** tasks while retaining RAG’s performance in factual QA.
> > >
> > > Using our comprehensive experimental design, we demonstrate that popular solutions like RAPTOR, GraphRAG and HippoRAG are still unable to endow LLMs with these abilities. **Only our method, ContinualRAG, is able to leverage the right set of technical innovations** —dense-sparse integration, deeper contextualization, and recognition memory—to achieve improved performance over standard RAG across the board.
> > >
> > > ## Synthetic Temporal Experiments
> > >
> > > As described above, our experiments are designed to test whether models can apply new knowledge in factual, associative, and sense-making tasks. However, we acknowledge that we initially did not assess catastrophic forgetting—a valuable point raised by reviewers.
> > >
> > > Existing continual learning QA benchmarks for LLMs (Liska et al. 2022, Kim et al. 2024) create different sets of documents based on their creation time and define three subsets: *unchanged*, *updated* and *new*. As the corpus evolves, performance on the *unchanged* subset reflects resistance to forgetting, while scores on the *updated* and *new* subsets measure learning of new information.
> > >
> > > Given that **no existing continual learning datasets test for associativity or sense-making**, we created synthetic temporal datasets from NQ and MuSiQue to evaluate these capabilities. Specifically, we randomly split each dataset into four sets and measure performance on one (our *unchanged* subset) as the other three *new* subsets were added. This allowed us to measure our method’s ability to **avoid catastrophic forgetting in both factual and associative scenarios**.
> > >
> > > ## References
> > >
> > > - Liska et al., StreamingQA: A Benchmark for Adaptation to New Knowledge over Time in QA Models. PMLR 2022
> > > - Kim et al., Carpe diem: On the Evaluation of World Knowledge in Lifelong Language Models. NAACL 2024
> > > - Gupta et al., Model Editing at Scale leads to Gradual and Catastrophic Forgetting. Findings of ACL 2024
> > > - Zhong et al., MQuAKE: Assessing Knowledge Editing in Language Models via Multi-Hop Questions. ACL 2023
> > > - Wang et al., DeepEdit: Knowledge Editing as Decoding with Constraints. 2024

---

### Official Review · Reviewer_gi53 · 2025-03-16

**Overall Recommendation:** 4

**Summary:**

**Main findings**:

The ability to continuously acquire, organize, and leverage knowledge is a fundamental aspect of human intelligence. To empower LLMs with this capability, retrieval-augmented generation (RAG) has emerged as a critical approach. Recent methods enhance vector embeddings by integrating structures such as knowledge graphs, aiming to improve sense-making and associativity. However, these advanced approaches often suffer from significant performance degradation on basic factual memory tasks compared to standard RAG.

**Main algorithmic/conceptual ideas**:

To address this issue, we propose **ContinualRAG**, which extends HippoRAG by introducing three key enhancements: **Dense-Sparse Integration**, **Deeper Contextualization**, and **Recognition Memory**.

**Main results**:

Experiments conducted on diverse benchmarks demonstrate that ContinualRAG outperforms previous state-of-the-art methods such as HippoRAG, GraphRAG, and NV-Embed-v2.

**Claims And Evidence:**

Yes, the claims are clear and convincing.

**Essential References Not Discussed:**

No additional related works should be included.

**Experimental Designs Or Analyses:**

Yes, I have checked the experimental designs.

**Methods And Evaluation Criteria:**

Yes, the proposed methods make sense for the target problems.

**Other Comments Or Suggestions:**

No additional comments.

**Other Strengths And Weaknesses:**

**Strengths**:

1. The proposed ContinualRAG is well-motivated and clearly explained.
2. ContinualRAG introduces solid technical improvements over HippoRAG.
3. ContinualRAG achieves impressively strong performance on various RAG benchmarks.

**Weakness**:

1. Given that ContinualRAG aims to empower LLMs to continuously acquire, organize, and leverage knowledge, additional experiments could be explored to continually update and expand the knowledge graph, which will make this work more solid.

**Questions For Authors:**

1. My suggestion is to consider conducting experiments that continuously introduce additional knowledge into the KG.

**Relation To Broader Scientific Literature:**

Introducing **ContinualRAG** can empower LLMs to continuously acquire, organize, and leverage knowledge in a human-like manner.

**Theoretical Claims:**

Not applicable. No theoretical claims.

---

> ### Author Rebuttal · Authors · 2025-03-31
>
> We sincerely appreciate the reviewer for the time and effort dedicated to reviewing our work. We are delighted that they found our work clear, convincing, well-motivated, technically solid and empirically validated by impressive performance improvements. We address their suggestions and comments below.
>
> ## Continual Knowledge Injection Experiments
>
> > Given that ContinualRAG aims to empower LLMs to continuously acquire, organize, and leverage knowledge, additional experiments could be explored to continually update and expand the knowledge graph, which will make this work more solid.
>
> We appreciate the reviewer’s thoughtful suggestion and agree with their assessment.
>
> To address this, we conduct a new experiment on both NQ and MuSiQue. We partition our full corpus into four equal segments (approximately 250 questions and their distractors). We then select one segment for evaluation and incrementally add the remaining segments, measuring how performance evolves as new knowledge is added—simulating a temporal continual learning setting. We report the performance of ContinualRAG and NV-Embed-V2, our strongest baseline, in the tables below using F1 scores.
>
> **NQ**
>
> | # of Documents | ContinualRAG| NV-Embed-V2 |
> |----------------|---------------|-----------------|
> | 5,171          | 60.83         | 60.26           |
> | 6,624          | 61.14         | 60.87           |
> | 8,098          | 61.35         | 60.66           |
> | 9,633          | 61.67         | 60.66           |
>
> **MuSiQue**
>
> | # of Documents | ContinualRAG | NV-Embed-V2 |
> |----------------|---------------|-----------------|
> | 3,316          | 52.83         | 47.44           |
> | 5,496          | 52.02         | 45.52           |
> | 8,127          | 49.93         | 44.13           |
> | 11,656         | 44.95         | 40.42           |
>
> As we can see, ContinualRAG’s improvements over NV-Embed-V2 remain remarkably consistent in both simple and associative continual learning settings. We note that, while both methods show steady performance on simple QA as more knowledge is introduced, their performance drops almost equivalently in the more complex task as more information is introduced. This behavior shows the strength of RAG in simple temporal tasks, highlights the value of our current experimental setup and points to the need for more complex temporal continual learning benchmarks for LLMs.
>
> We will add this experiment to the camera-ready version of our paper.

---

> > ### Comment · Reviewer_gi53 · 2025-04-04
> >
> > The rebuttal solves my concerns very well. I keep my rating of accept. Great work!

---

### Decision · Program_Chairs · 2025-05-01

**Decision:**

Accept (poster)

**Comment:**

This paper presents a method to enhance traditional RAG with a continual learning framework for large language models. While all the reviewers agree that the problem is well-motivated and the solution is intuitive and effective, there are some aspects that can be improved  and integrated into the next version of this work: i) more careful comparison with HippoRAG; ii) experiments emphasizing document storage; iii) experiments that incorporate a temporal dimension.